# Mitochondrial stress response triggered by defects in protein synthesis quality control

Uwe Richter[1],* , Kah Ying Ng[1],* , Fumi Suomi[1],*, Paula Marttinen[1], Taina Turunen[1], Christopher Jackson[2], Anu Suomalainen[2], Helena Vihinen[1], Eija Jokitalo[1], Tuula A Nyman[3] , Marita A Isokallio[4], James B Stewart[4] , Cecilia Mancini[5], Alfredo Brusco[5] , Sara Seneca[6], Anne Lombès[7] , Robert W Taylor[8], Brendan J Battersby[1]

**Mitochondria have a compartmentalized gene expression system dedicated to the synthesis of membrane proteins essential for oxidative phosphorylation. Responsive quality control mechanisms are needed to ensure that aberrant protein synthesis does not disrupt mitochondrial function. Pathogenic mutations that impede the function of the mitochondrial matrix quality control protease complex composed of AFG3L2 and paraplegin cause a multifaceted clinical syndrome. At the cell and molecular level, defects to this quality control complex are defined by impairment to mitochondrial form and function. Here, we establish the etiology of these phenotypes. We show how disruptions to the quality control of mitochondrial protein synthesis trigger a sequential stress response characterized first by OMA1 activation followed by loss of mitochondrial ribosomes and by remodelling of mitochondrial inner membrane ultrastructure. Inhibiting mitochondrial protein synthesis with chloramphenicol completely blocks this stress response. Together, our data establish a mechanism linking major cell biological phenotypes of AFG3L2 pathogenesis and show how modulation of mitochondrial protein synthesis can exert a beneficial effect on organelle homeostasis.**

## Introduction

Mitochondria contain a unique protein synthesis machinery devoted to the exclusive translation of a small number of proteins encoded in the mitochondrial genome (Ott et al, 2016). In humans, these number only 13 hydrophobic membrane proteins, which form core subunits of three respiratory chain complexes and the $F_1F_O$ ATP synthase required for oxidative phosphorylation. Assembly of these complexes requires an additional 80 structural proteins encoded in the nucleus and imported into mitochondria, so protein synthesis in the cytosol and mitochondria needs temporal and spatial coordination. On the mitochondrial side, protein synthesis must be tightly regulated with complex assembly or proteolytic quality control to prevent nascent chains from over-accumulating in the membrane.

Proteostasis within mitochondria requires a collection of compartmentalized chaperones and proteases that trace their evolutionary origin back to alphaproteobacteria (Quirós et al, 2015). One factor implicated in the quality control of mitochondrial protein synthesis is a membrane-anchored AAA (ATPases Associated with diverse cellular Activities) protease complex composed of AFG3L2 subunits. In humans, this hexameric complex affects the stability of newly synthesized mitochondrial proteins (Zurita Rendón & Shoubridge, 2012; Hornig-Do et al, 2012; Richter et al, 2015). In the budding yeast, proteolytic maturation of the mitochondrial ribosomal protein Mrpl32 has been ascribed to this AAA protease complex, a step required for ribosome assembly (Nolden et al, 2005). In contrast, this mechanism is not observed in mammalian tissues (Almajan et al, 2012). It is worth noting that site-specific proteolytic processing of substrates appears to be incompatible with the conserved mechanistic function of AAA proteases, which couple ATP hydrolysis for protein unfolding with processive proteolysis of substrates into oligopeptides (10 amino acids) (Sauer & Baker, 2011). Thus, the effect on ribosome biogenesis is possibly indirect, perhaps, reflecting a downstream response of mitochondrial dysfunction.

Pathogenic mutations that disrupt the AFG3L2 quality control complex are associated with human diseases: hereditary spastic paraplegia (HSP), spastic ataxia (SPAX5), spinocerebellar ataxia (SCA28), and progressive external ophthalmoplegia (PEO) (Casari et al, 1998; Di Bella et al, 2010; Pierson et al, 2011; Gorman et al, 2015). A hallmark of these diseases, also seen in mouse models (Almajan et al, 2012), is fragmentation of the mitochondrial membrane

---

[1]Institute of Biotechnology, University of Helsinki, Helsinki, Finland    [2]Research Programs Unit—Molecular Neurology, University of Helsinki, Helsinki, Finland [3]Department of Immunology, Institute of Clinical Medicine, University of Oslo and Oslo University Hospital, Oslo, Norway    [4]Max Planck Institute for Biology of Ageing, Cologne, Germany    [5]Department of Medical Sciences, University of Torino, Torino, Italy    [6]Center for Medical Genetics/Research Center Reproduction and Genetics, Universitair Ziekenhuis Brussel, Brussels, Belgium    [7]Faculté de médecine Cochin, Institut Cochin Institut national de la santé et de la recherche médicale U1016, Centre national de la recherche scientifique Unités Mixtes de Recherche 8104, Université Paris 5, Paris, France    [8]Wellcome Centre for Mitochondrial Research, Institute of Neuroscience, Newcastle University, Newcastle upon Tyne, UK

Correspondence: brendan.battersby@helsinki.fi
*Uwe Richter, Kah Ying Ng, and Fumi Suomi contributed equally to this work.

morphology, loss of mitochondrial ribosomes, and defective oxidative phosphorylation. Fragmentation of mitochondrial morphology prevents axonal trafficking in neurons (Kondadi et al, 2014) and is considered a key cell biological event in the molecular pathogenesis of these disorders (Ferreirinha et al, 2004). Nonetheless, the etiology of these three distinguishing phenotypes has remained unknown for the last 20 y.

AFG3L2 dysfunction is known to activate the metalloprotease OMA1, triggering processing of the dynamin-related GTPase OPA1 tethered to the inner membrane releasing it into a soluble pool (Ehses et al, 2009). This activation mechanism appears to be independent of the membrane potential (Ehses et al, 2009). Under these circumstances, mitochondrial morphology is remodelled from a reticular shape to a fragmented swollen state. Preventing stress-induced OPA1 processing has been shown to be beneficial in a mouse model of neurodegeneration (Korwitz et al, 2016). However, no mechanism has been established, so far, that could reconcile AFG3L2 quality control to OPA1 processing and dynamic changes in membrane morphology.

In this study, we report that impaired quality control of mitochondrial nascent chain synthesis triggers a novel stress response first characterized by OMA1 activation and then remodelling of membrane ultrastructure from the reduction in mitochondrial ribosomes. Together, our data establish a mechanism linking major cell biological phenotypes of AFG3L2 pathogenesis and show how modulation of mitochondrial protein synthesis can exert a beneficial effect on organelle homeostasis.

## Results

### A mitochondrial stress response from impaired nascent chain quality control

AFG3L2 dysfunction activates OMA1, inducing proteolysis of the long OPA1 forms (membrane tethered) independently of alterations to membrane potential (Ehses et al, 2009; Kondadi et al, 2014) (Fig 1A–C). Recently, we showed how AFG3L2 modulated a mitochondrial membrane proteotoxicity arising from the small molecule actinonin and characterized by OMA1 activation (Richter et al, 2013, 2015). The magnitude of the drug response was proportional to the rate of translation elongation on mitochondrial ribosomes and/or the abundance of AFG3L2. A subsequent study demonstrated actinonin specifically bound to the AFG3L2 orthologue in *Plasmodium falciparum* and *Toxoplasma gondii* (Amberg-Johnson et al, 2017), suggesting a conserved mechanism across the eukaryotic tree. Therefore, we asked whether the OMA1 activation associated with AFG3L2 defects was triggered by stress arising from mitochondrial protein synthesis. *AFG3L2* siRNA leads to proteolytic cleavage of long OPA1 and mitochondrial fragmentation (Fig 1B and D). Both phenotypes were robustly blocked by inhibiting protein synthesis on mitochondrial ribosomes with chloramphenicol (Fig 1B and D).

Another prominent phenotype observed with AFG3L2 dysfunction in genetically modified mouse models is reduction in the abundance of mitochondrial ribosomes (Nolden et al, 2005; Almajan et al, 2012). After 5 d of *AFG3L2* siRNA knockdown in human

cultured fibroblasts, there was no difference on the abundance of the mitochondrial ribosomal protein uL11m (Fig 1B). This suggests the ribosomal phenotype may not be direct but rather a response to the progressive loss of this AAA protease complex. To test the idea, we investigated the consequences from a longer *AFG3L2* siRNA knockdown by waiting an additional three days (no extra siRNA was added) before collecting the cells. The rationale for this approach is that *AFG3L2* is a core fitness gene (Hart et al, 2015), which impedes the use of genome editing to make a genetic knockout in cultured human cells. After 8 d of culture, we still observed a robust reduction of AFG3L2 and OPA1 processing, which is now accompanied with a decrease in mitochondrial ribosomal proteins from both the large and small subunit (Fig 1B). Inhibition of mitochondrial protein synthesis with chloramphenicol completely suppressed not only OPA1 processing but also the reduction of mitochondrial ribosomes (Fig 1B). Together, the data indicate AFG3L2 dysfunction generated a proteotoxicity that arose specifically from newly synthesized mitochondrial proteins, triggering an organelle stress response.

Because the mitochondrial genome (mtDNA) encodes both ribosomal RNAs (rRNAs) essential for mitochondrial ribosome assembly, we sought to distinguish whether the loss of the mitochondrial ribosomes was simply a downstream response from a reduced mtDNA copy number. OPA1 abundance is known to affect the copy number and integrity of the mitochondrial genome (mtDNA) (Hudson et al, 2008). We used Southern blotting to quantify the mitochondrial copy number and genome integrity. No differences were observed with either short- or long-term *AFG3L2* siRNA knockdown (Fig 1E and F).

Prohibitins form a scaffold in the inner membrane that physically interacts with and regulates the AFG3L2 complex (Steglich et al, 1999; Nolden et al, 2005). Loss of the prohibitin scaffold also triggers OMA1-dependent processing of OPA1 (Merkwirth et al, 2008). To test if mitochondrial nascent chain proteotoxicity was the OMA1 trigger, we used siRNA against prohibitin 2 (*PHB2*) and treated cells with chloramphenicol. Complete inhibition of mitochondrial protein synthesis with chloramphenicol blocked the OPA1 processing (Fig S1A). In contrast, loss of the other mitochondrial inner membrane–bound AAA protease, YME1L, or its scaffold STOML2 (Wai et al, 2016) did not induce a translation-dependent OPA1 processing (Fig S1B and C). This points to a primary importance of the AFG3L2 complex for the quality control of nascent chains emerging from mitochondrial ribosomes in the matrix.

Next, we addressed the relevance of the stress response with a pathogenic mutation in AFG3L2. MEFs heterozygous for a knock-in mutation M665R (M666R in humans) associated with spinocerebellar ataxia 28 (SCA28) (Cagnoli et al, 2010; Mancini et al, 2018) did not exhibit stress-activated OPA1 proteolysis or reduction of mitochondrial ribosomal proteins at the steady state (Fig 1G). This finding is consistent with observations from a number of human heterozygous pathogenic mutations in AFG3L2 (Di Bella et al, 2010). In contrast, MEFs homozygous for the M665R mutation exhibited mitochondrial translation-dependent OPA1 processing at the steady state (Fig 1G).

Temperature is a well-known modulator of protein folding, so we used heat shock as a stress during mitochondrial protein synthesis to test whether it is a modifier of the M665R Afg3l2 mutation. Heat shock triggered mitochondrial translation-dependent OPA1 processing and reduction of mitochondrial ribosomal proteins

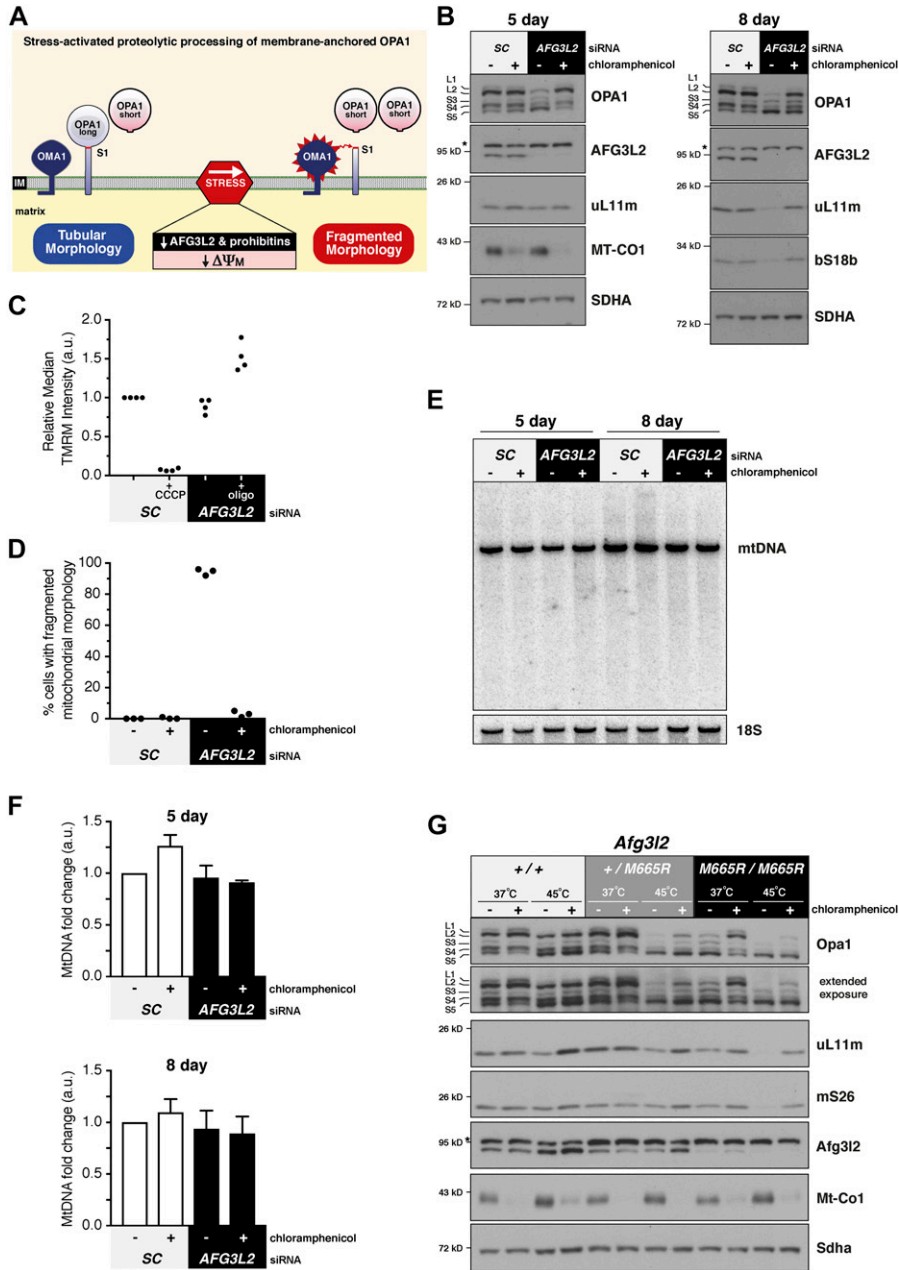

**Figure 1. A mitochondrial translation stress response regulating OPA1 and ribosome homeostasis.**
**(A)** Diagram illustrating the stress-induced regulated proteolysis of membrane-tethered OPA1 by the OMA1 metalloprotease. **(B)** Immunoblotting of whole-cell lysates of human fibroblasts treated with the indicated siRNAs with (+) and without (−) chloramphenicol (CA). *Nonspecific band detected with the AFG3L2 antibody. **(C)** Scatter plot of mitochondrial membrane potential analyzed by tetramethylrhodamine (TMRM) staining of HEK293 cells treated with the indicated siRNAs from independent experiments. CCCP is a positive control to dissipate the membrane potential. Oligomycin (oligo) inhibits the F1FO ATP synthase, which leads to hyperpolarization unless ATP hydrolysis is required to maintain the membrane potential. **(D)** Scatter plot quantification of mitochondrial morphology under light microscopy in human fibroblasts treated with the indicated siRNAs with or without CA. Data are from three independent experiments, n = 100. **(E)** Southern blot of human fibroblasts treated with the indicated siRNAs with (+) and without (−) CA. **(F)** Quantification of Southern blotting in (E) from three independent experiments. Data represent mean ± SD. **(G)** Immunoblotting of mouse embryonic fibroblasts with the indicated *Afg3l2* genotypes grown at 37°C or heat shocked for 4 h at 45°C with (+) and without (−) CA. IM, inner membrane; S1, OMA1 proteolytic site; SC, scrambled.

proportional to the allelic copy number of the pathogenic M665R Afg3l2 mutation (Fig 1G). This effect was not observed in the wild-type MEFs. To investigate the robustness of the finding in further detail, we used heat shock in a series of control cell lines (wild-type human immortalized fibroblasts, myoblasts, HEK293, and an additional MEF line) for up to 4 h. None of these human or mouse cultured cell lines exhibited stress-activated OPA1 processing with heat shock (Figs S1C–E and S2D and G). Thus, heat shock appeared to elicit a mitochondrial translation-dependent stress response only with pathogenic mutations in AFG3L2. Collectively, our data demonstrate that AFG3L2 has a primary role in regulating proteotoxicity that arises during protein synthesis on mitochondrial

ribosomes. Absence of this quality control step activates a conserved progressive mitochondrial stress response initiated on the inner membrane.

## Aberrant mitochondrial mRNA triggers a translation-dependent stress response

Next, we set out to investigate the source of the proteotoxicity that arises during nascent chain synthesis on mitochondrial ribosomes. Proteotoxicity could be generated by the impaired quality control of all 13 proteins, a subset or specific nascent chains, or a type/class of genetic mutation. Currently, it is not possible to modify open

reading frames within the human mitochondrial genome with gene editing, so we turned to genetic mutations linked to human mitochondrial disorders that disrupt the synthesis or stability of the 13 mitochondrial proteins.

Our first goal was to test whether unstable nascent chains triggered OMA1 activation. We took several approaches to investigate this question, focusing on critical subunits required for the assembly of individual oxidative phosphorylation complexes. Assembly defects to the $F_1F_O$ ATP synthase generate instability of MT-ATP6 and MT-ATP8 nascent chains (Fig S2A and B). We tested a mild and severe disruption to complex assembly (Čížková et al, 2008; Rak et al, 2011), but neither induced OMA1 activation (Fig S2C). Next, we turned to the two catalytic core subunits of cytochrome c oxidase: MT-CO1 and MT-CO2. COX10 is an essential enzyme involved in the biosynthetic pathway for heme *a*, a critical metal moiety for cytochrome c oxidase assembly and required for stability of MT-CO1 nascent chains (Antonicka et al, 2003). The absence of MT-CO1 also disrupts MT-CO2 stability (Fig S2D). Genetic disruptions to *COX10* function, however, did not activate OMA1 even after heat shock (Fig S2E). These data suggest that instability of four of the 13 nascent chains does not act as proteotoxic trigger.

To expand this line of inquiry, we asked whether a severe disruption to the synthesis and stability of all 13 nascent chains could be an OMA1 trigger. To test this question, we used human myoblasts homoplasmic for the m.8344A>G mutation in the mitochondrial tRNA[Lys] linked to MERRF (myoclonus epilepsy, ragged-red fibers) (Boulet et al, 1992; Sasarman et al, 2008). In these cells, there is a severe defect to the synthesis and stability of all mitochondrial nascent chains (Fig S2F and G) (Richter et al, 2018). However, we observed no stress-induced OMA1 activation at the steady-state or even following a 4-h heat shock (Fig S2G). Collectively, our data rule out the instability of mitochondrial nascent chains as the proteotoxic trigger for OMA1 activation.

Most human pathogenic mtDNA mutations occur within tRNA genes and tend to be very rare in the protein-coding regions (Taylor & Turnbull, 2005; Yarham et al, 2010). An exception is *MT-ATP6*, which is an essential subunit of the $F_1F_O$ ATP synthase (Zhou et al, 2015). We investigated whether mutations in this open reading frame could induce the mitochondrial translation-dependent stress response. In this collection of patient fibroblasts, there are three different missense mutations that generate apparently stable proteins at steady state, one nonsense mutation that leads to reduced synthesis of the full-length MT-ATP6, and an *MT-ATP6* mRNA lacking a stop codon (Fig 2A and B) (Seneca et al, 1996; Auré et al, 2013; Jackson et al, 2017). At steady state, none of these pathogenic mutations displayed stress-induced OPA1 processing

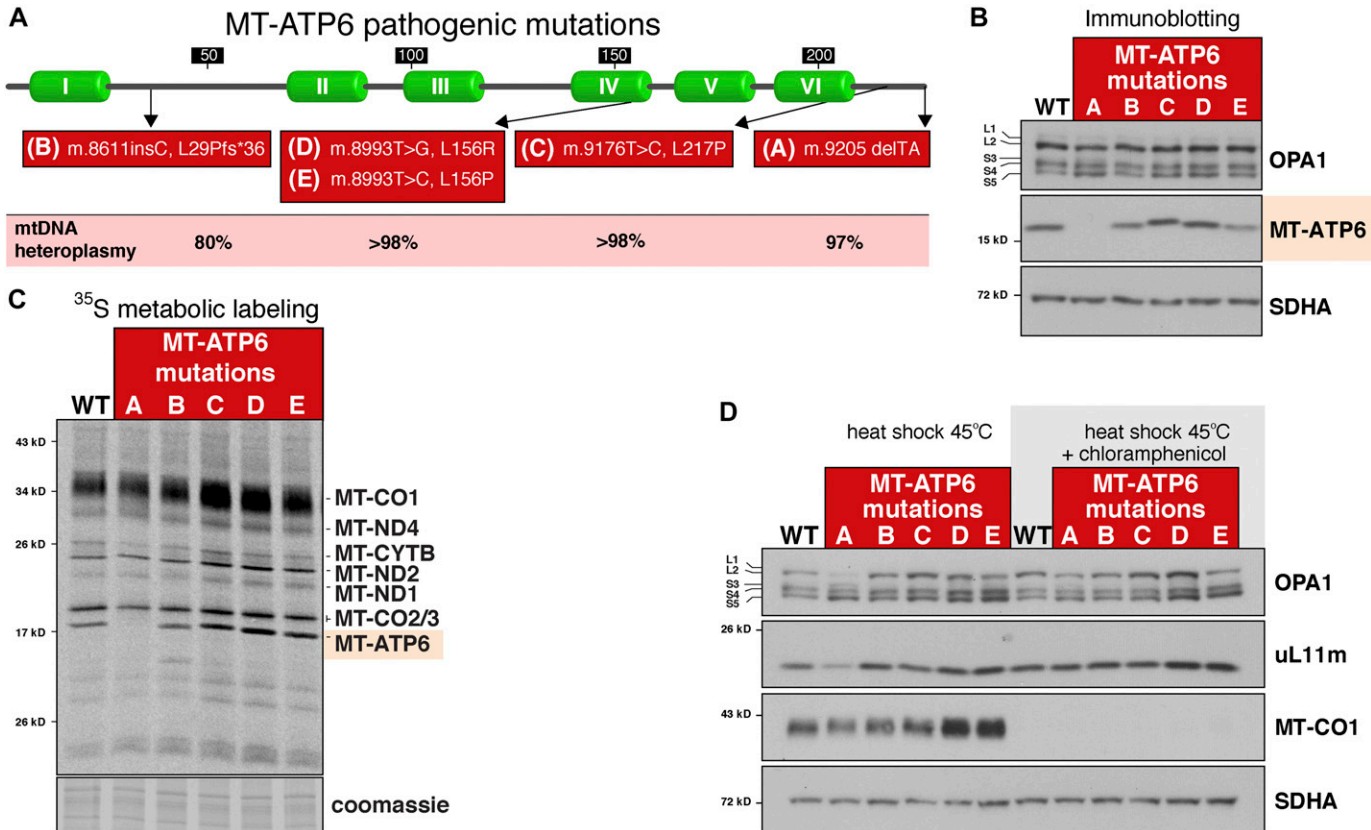

**Figure 2. Heat shock during translation of an aberrant mitochondrial mRNA triggers a mitochondrial stress response.**
**(A)** Diagram of human MT-ATP6 illustrating the location of pathogenic mutations in the protein and the heteroplasmy level of the mutations in patient fibroblasts. **(B)** Immunoblotting of whole-cell lysates from patient fibroblasts with the indicated *MT-ATP6* mutations and a WT control. **(C)** A 30-min pulse metabolic labelling with [35]S-methionine/cysteine of mitochondrial protein synthesis in patient fibroblasts with the indicated *MT-ATP6* mutations. **(D)** Immunoblotting of whole-cell lysates from fibroblasts with the indicated *MT-ATP6* mutations heat shocked for 4 h at 45°C with (+) and without (−) chloramphenicol.

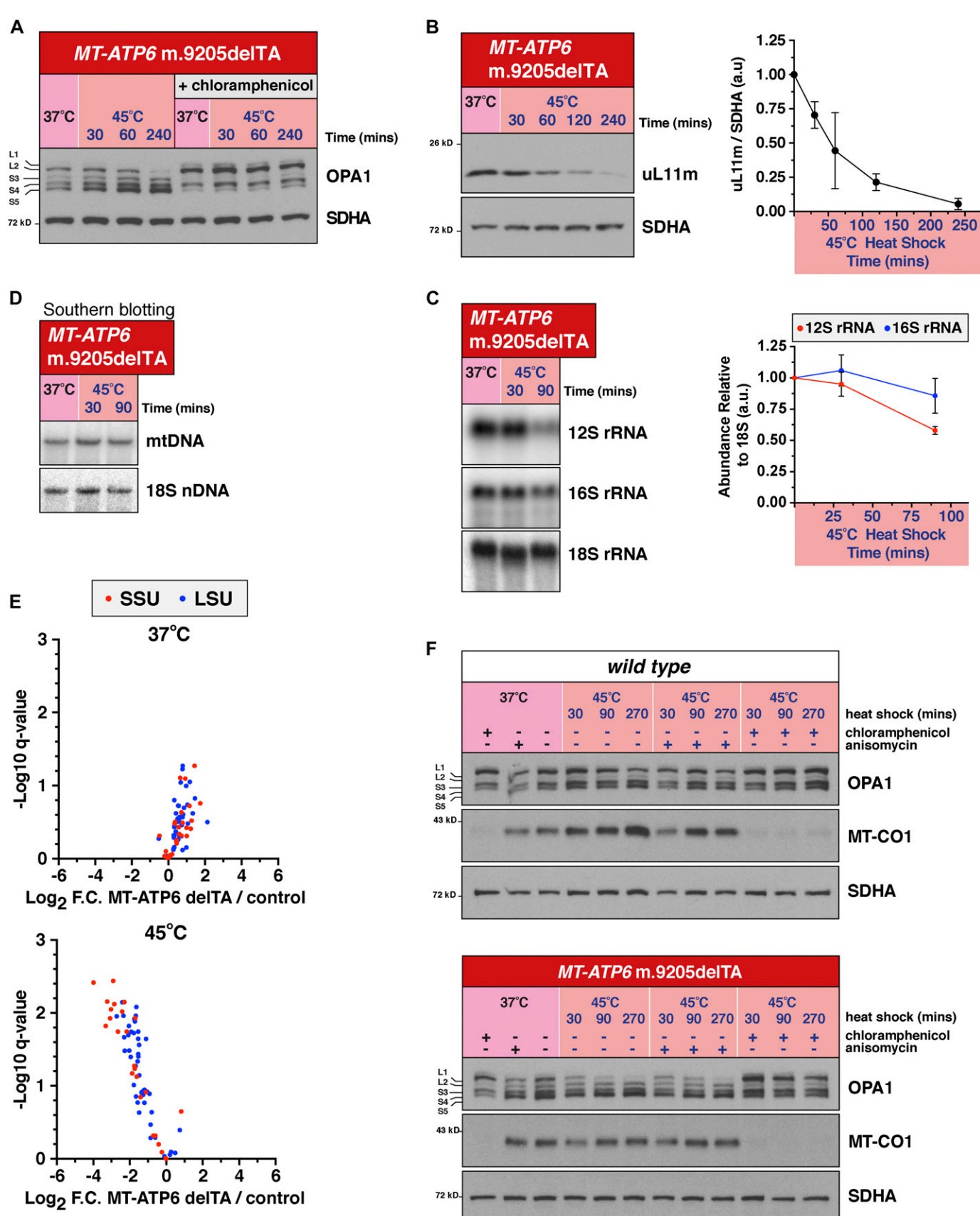

(Fig 2B). The non-stop *MT-ATP6* mRNA (m.9205delTA), because of a 2-bp microdeletion, was associated with undetectable protein synthesis (Fig 2B and C). Using high-resolution respirometry, however, along with an oligomycin titration assay, which is a specific enzymatic inhibitor of the $F_1F_O$ ATP synthase, we could measure a residual level of a functional complex (Fig S3). Deep sequencing of the mitochondrial genomes in the *MT-ATP6* m.9205delTA fibroblasts (Table S1) identified approximately 3% wild-type mtDNA molecules, which is presumably the source for the functional MT-ATP6 that was undetectable by metabolic labelling or immunoblotting.

We asked whether heat shock could modulate any of the *MT-ATP6* mutations and thereby activate the mitochondrial translation stress response. To our surprise, heat shock induced OPA1 proteolytic processing but only with the *MT-ATP6* m.9205delTA mutation (Fig 2D). Heat shock also reduced the abundance of the mitochondrial ribosomal protein uL11m in these fibroblasts (Fig 2D). Both phenotypes could be completely blocked when mitochondrial protein synthesis was inhibited with chloramphenicol (Fig 2D). Together, the data suggested that only nascent chain synthesis from an aberrant mitochondrial mRNA during heat shock can activate the same stress response observed with AFG3L2 dysfunction.

Next, we sought to investigate in more detail the mitochondrial translation-dependent stress response with the *MT-ATP6* m.9205delTA mutation. Heat shock induced time-dependent processing of OPA1 and reduction in uL11m (Fig 3A and B). Interestingly, there is a differential decay on the reduction in the mitochondrial rRNA, which appears to be faster for the 12S rRNA of the small subunit (Fig 3C). Importantly, this loss of mitochondrial rRNA and ribosomal protein is not due to alterations in the copy number of mtDNA (Fig 3D). To generate a more comprehensive and quantitative analysis of the changes to mitochondrial ribosomal protein abundance, we used label-free quantitative liquid chromatography–mass spectrometry/mass spectrometry. Using this method, we detected a significant and robust loss of mitochondrial ribosomal proteins from both the large and small subunits during heat shock in the *MT-ATP6* m.9205delTA patient fibroblasts (Fig 3E). Assembly of the $F_1F_O$ ATP synthase requires subunits synthesized on both cytosolic and mitochondrial ribosomes. Thus, unbalanced accumulation of newly synthesized subunits of the complex from cytosolic ribosomes could trigger the response in the *MT-ATP6* m.9205delTA patient fibroblasts during heat shock. To test this idea, we inhibited cytoplasmic protein synthesis with anisomycin, a well-known inhibitor of the 80S cytosolic ribosome but did not observe any modulation of the OPA1 processing phenotype (Fig 3F). Therefore, heat shock during the translation of an aberrant mitochondrial mRNA triggers a ribosomal decay pathway and OMA1 activation.

To address the consequence of heat shock to mitochondrial protein synthesis, we pulse labelled mitochondrial proteins with [35]S for 30 min at different time points during a 4-h incubation (Fig 4A). In the *MT-ATP6* m.9205delTA fibroblasts, we observed a rapid attenuation of mitochondrial protein synthesis during heat shock (Fig 4B) that preceded the reduction in ribosomal proteins (Fig 3B). Because we showed AFG3L2 is integral to the quality control of mitochondrial nascent chain synthesis (Fig 1), we asked whether a quantitative increase in the complex abundance could modulate the stress response in the *MT-ATP6* m.9205delTA fibroblasts. Stable retroviral overexpression of the wild-type *AFG3L2* cDNA in these fibroblasts suppressed both the OPA1 processing and mitochondrial ribosomal phenotypes (Fig 4C). Because the complex can act as an AAA unfoldase independently of its role in proteolysis (Arlt et al, 1996), we sought to distinguish which function was most important during heat shock. To this end, we used site-directed mutagenesis to insert an E575Q mutation in the HEXXH motif of the AFG3L2 protease domain, although the protein still retains a functional AAA domain. A quantitative increase in the AAA function of AFG3L2 was epistatic to the translation of the m.9205delTA mutation during heat shock, suppressing both the OPA1 and the mitochondrial ribosomal phenotypes indistinguishably from the wild-type cDNA (Fig 4C). In contrast, AFG3L2 overexpression had no effect on the rapid attenuation of mitochondrial protein synthesis during heat shock (Fig 4D). Taken together, our findings revealed proteotoxicity arising from the translation of an aberrant mitochondrial mRNA and AFG3L2 dysfunction. The ability of the AFG3L2 complex to unfold these nascent chain substrates appears to be a critical modulator of this proteotoxicity arising from mitochondrial gene expression.

## Remodelling of lamellar cristae due to inhibition of mitochondrial protein synthesis

The OPA1 stress-induced proteolytic processing associated with AFG3L2 dysfunction has been linked to alterations in cristae morphology (Merkwirth et al, 2008; Ehses et al, 2009; MacVicar & Langer, 2016). However, our data clearly demonstrated this mitochondrial quality control step is also integrated with mitochondrial gene expression. Because the respiratory chain complexes and the $F_1F_O$ ATP synthase are localized within cristae (Rabl et al, 2009), biogenesis of these membrane invaginations is predicted to be coupled to the synthesis and assembly of the oxidative phosphorylation complexes. Therefore, we sought to distinguish primary versus potential feedback responses in cristae morphogenesis among these various factors. The ability to block OPA1 stress-activated

**Figure 3. Heat shock during translation of an aberrant mitochondrial mRNA activates OMA1 and a ribosome decay pathway.**
**(A)** Immunoblotting of whole-cell lysates from the *MT-ATP6* m.9205delTA fibroblasts. **(B)** Left, immunoblotting of whole-cell lysates from the *MT-ATP6* m.9205delTA fibroblasts. Right, quantification of the immunoblotting from three independent experiments. Data represent mean ± SD. **(C)** Left, representative images of Northern blotting of total cellular RNA from *MT-ATP6* m.9205delTA fibroblasts. Right, quantification of the Northern blotting from four independent experiments. Data represent mean ± SD. **(D)** A representative image of Southern blotting for mitochondrial DNA copy number from the *MT-ATP6* m.9205delTA fibroblasts from multiple independent experiments. **(E)** Quantitative label-free liquid chromatography–mass spectrometry/mass spectrometry analysis of mitochondrial ribosomal subunits from the large (LSU) and small (SSU) subunit of wild-type and *MT-ATP6* m.9205delTA fibroblasts grown at 37°C or heat shocked for 4 h at 45°C. Each data point represents a single mitochondrial ribosomal protein. Data were collected from isolated mitochondria from five independent experiments for each genotype and temperature condition. **(F)** Immunoblotting of whole-cell lysates of human fibroblasts with the indicated *MT-ATP6* genotypes following culture at 37°C or heat shocked for the indicated time at 45°C treated with the indicated translation inhibitors. Anisomycin inhibits translation elongation on cytoplasmic ribosomes.

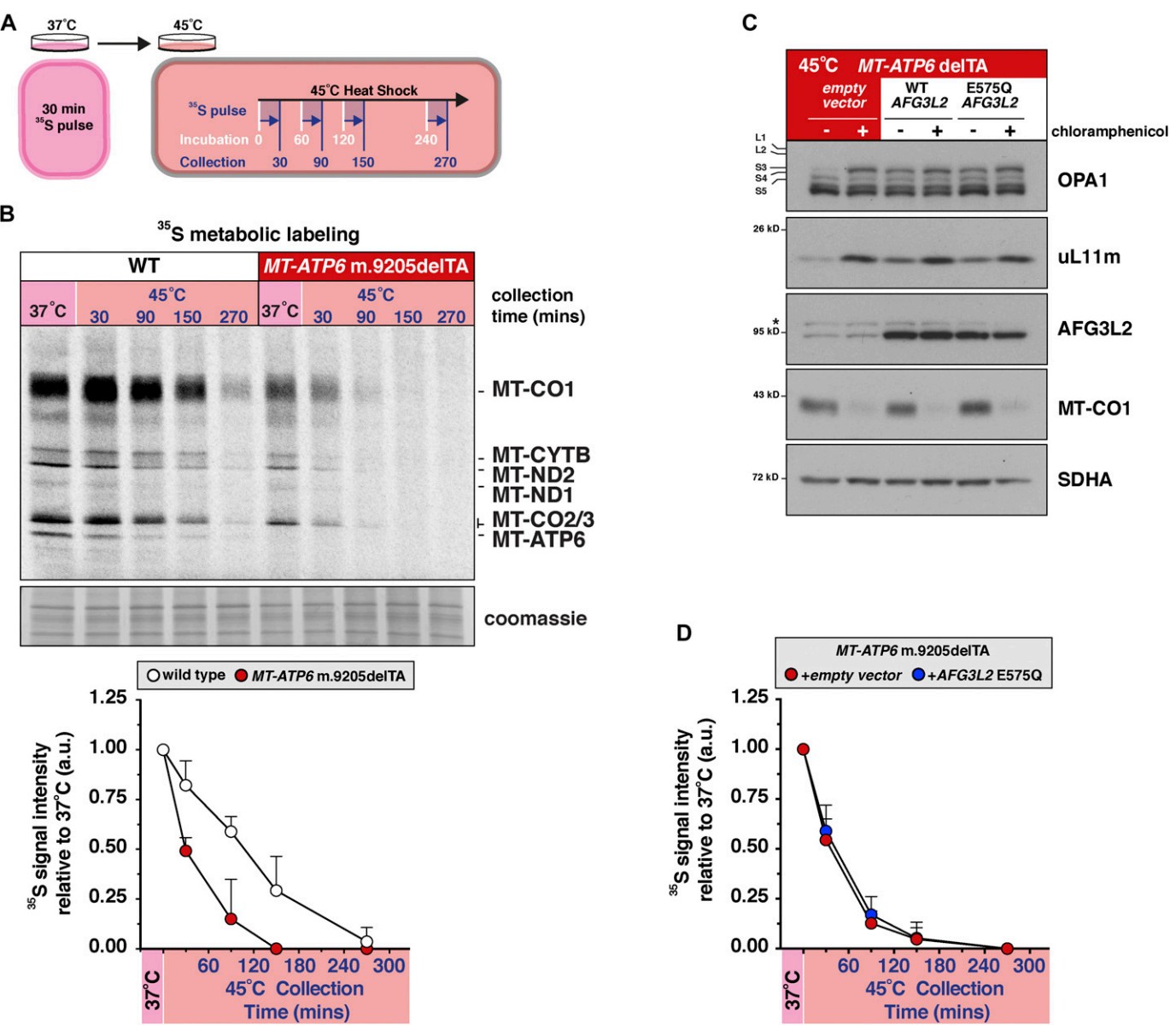

**Figure 4. Translation of an aberrant mitochondrial mRNA negatively regulates protein synthesis during heat shock.**
**(A)** A schematic illustrating the experimental workflow for ³⁵S-methionine/cysteine metabolic labelling for mitochondrial protein synthesis during heat shock. All cells were pulse labelled with ³⁵S-methionine/cysteine for 30 min to measure mitochondrial protein synthesis following the indicated incubation times at 45°C. **(B)** Top, a representative metabolic labelling with ³⁵S-methionine/cysteine during heat shock in human fibroblasts with the indicated *MT-ATP6* genotypes. Below, quantification of ³⁵S-incorporation into mitochondrial proteins during heat shock. Data represent mean + SD from four independent experiments. a.u., arbitrary units. **(C)** Human fibroblasts with the *MT-ATP6* m.9205delTA genotype were stably transduced by retrovirus with an empty vector, wild-type *AFG3L2* or *E575Q AFG3L2* cDNA. Immunoblotting of whole-cell lysates from cells heat shocked for 4 h at 45°C. *Nonspecific band detected with the AFG3L2 antibody. **(D)** Quantification of ³⁵S-incorporation into mitochondrial proteins in human fibroblasts with the *MT-ATP6* m.9205delTA genotype (from C). Data represent mean ± SD from four independent experiments. All data are representative of multiple independent experiments.

processing with chloramphenicol (Figs 1, 2, 3, and 4) provided a robust tool to investigate the question.

In human fibroblasts, mitochondrial cristae appear as lamellar sheets with single or multiple crista junctions, either circular or slot-like in shape, when imaged in situ with electron tomography and 3D modelling (Fig 5A–C, Video 1). Dimerization of the F₁FO ATP synthase generates positive curvature along cristae (Rabl et al, 2009; Davies et al, 2012). However, only with a severe assembly defect in the complex (e.g.,

*ATP5B* siRNA) did we observe a robust alteration in cristae morphology in transmission electron micrographs (Figs S2A and B, S3, and S4A), which appears similar to that seen with dimerisation defects (Rabl et al, 2009; Harner et al, 2016). In contrast, milder assembly defects or pathogenic *MT-ATP6* mutations associated with human disease had lamellar cristae (Figs 5A–C, S2A and B, S3, S4B–D, and Video 1 and 2). Apparently, a low level of a functional F₁FO ATP synthase (Fig S3) is sufficient to maintain cristae curvature (Figs 5A and C, S4, and Video 1 and 2).

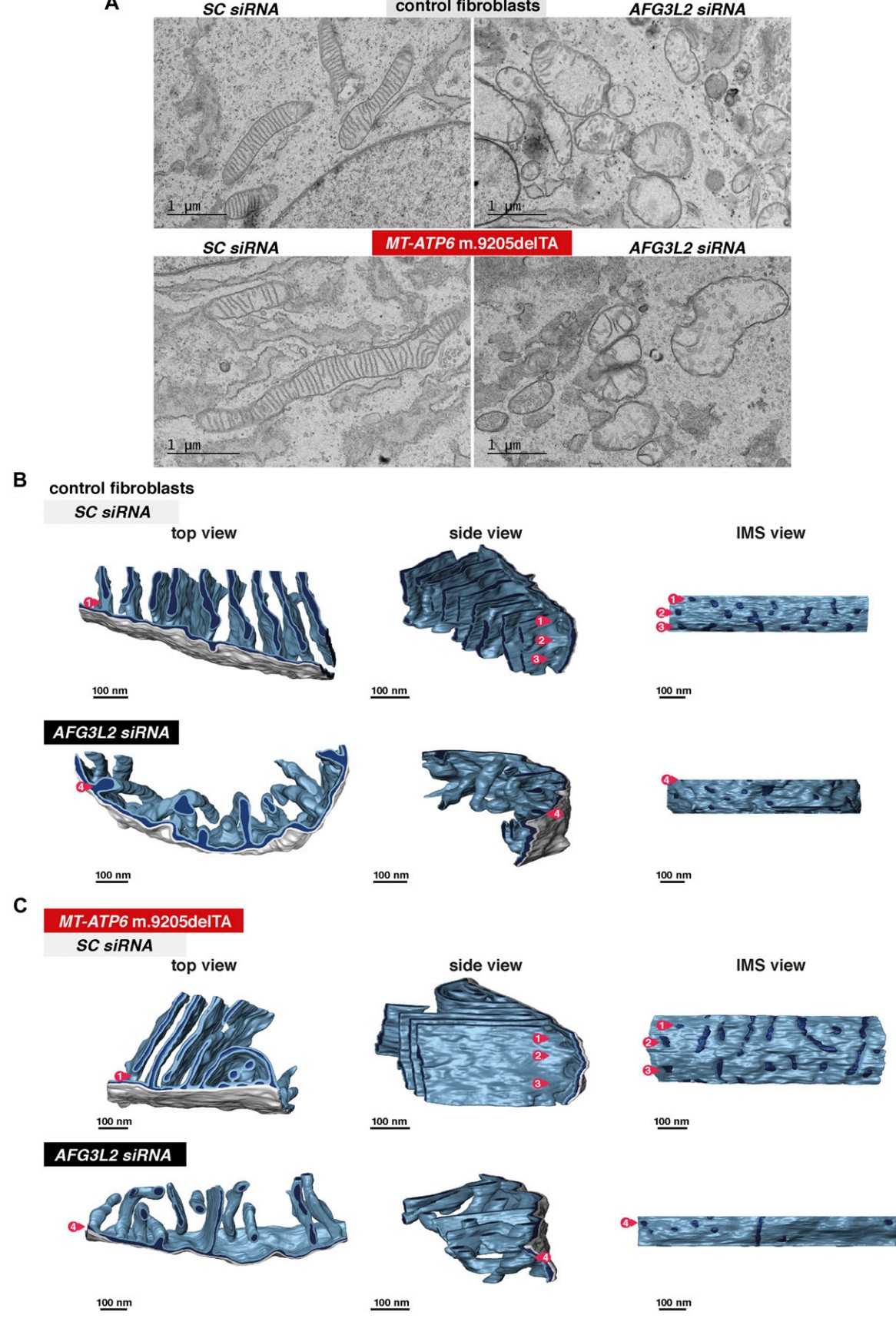

In contrast, *AFG3L2* siRNA induced mitochondrial swelling and disrupted the ordered lamellar ultrastructure: sheets collapsed, appearing as swollen tubes (Fig 5A–C and Video 3 and 4). Of note, tubular cristae still had intact circular junctions indistinguishable from those of wild-type (Fig 5B and C) even with a robust loss of the long membrane-tethered OPA1 forms (Fig 1B). Chloramphenicol blocks the stress-induced OPA1 processing and mitochondrial fragmentation associated with *AFG3L2* dysfunction (Fig 1B and D), but yet, we still observed remodelling of the inner membrane into short stubby cristae tubules and rings (Fig 6A). In fact, the inhibition of mitochondrial protein synthesis with chloramphenicol alone in wild-type fibroblasts dramatically remodelled the inner membrane without organelle fragmentation (Figs 1D and 6A). Short stubby cristae tubules were not observed with specific defects to the assembly of the $F_1F_O$ ATP synthase (Fig S4). Thus, a robust alteration in cristae morphogenesis appeared to arise with a global reduction in mitochondrial protein synthesis independent of OPA1 processing. This remodelling event could possibly serve as a feedback response to membrane biogenesis from defective assembly of all four oxidative phosphorylation complexes (I, III, IV, and V).

To further test this hypothesis, we examined the mitochondrial ultrastructure in two established cases with severe combined oxidative phosphorylation deficiencies but lacking stress-induced OPA1 processing: the mtDNA m.8344A>G tRNA$^{Lys}$ mutation and *C12orf65* deficiency. In myoblasts homoplasmic for the m.8344A>G tRNA$^{Lys}$ mutation, we observed short stubby cristae that appeared as tubules and not lamellar sheets compared with myoblasts with the same nuclear background but wild-type mtDNA (Fig 6B). Furthermore, in human fibroblasts with *C12orf65* deficiency (Antonicka et al, 2010), we also observed a similar cristae phenotype that was restored to lamellar sheets following functional complementation with retroviral expression of the wild-type *C12orf65* cDNA (Fig 6C).

Last, we addressed how heat shock to the *MT-ATP6* m.9205delTA fibroblasts affected mitochondrial morphology and ultrastructure. A 4-h heat shock did not induce mitochondrial fragmentation or alterations in the inner membrane ultrastructure (Fig 6D and E) even with robust stress-induced OPA1 processing (Figs 2D, 3A and F, and 4C). This would indicate that additional signals to the actual proteolytic processing of OPA1 are required during heat shock to trigger mitochondrial fragmentation. Our findings indicate how mitochondrial protein synthesis per se is a major functional determinant for cristae morphogenesis. Thus, any organelle stress response that impinges globally upon mitochondrial gene expression will initiate remodelling of the inner membrane ultrastructure.

## Discussion

Proteotoxicity has long been considered a key factor in the molecular pathogenesis of mitochondrial dysfunction in human disease, but the sources and origin of the offending toxic substrates have remained unclear. Here, we demonstrated how disruptions to mitochondrial nascent chain quality control act as a proteotoxic trigger, initiating a stress response that links inner membrane ultrastructure and ribosome homeostasis. These are key features observed at the cell biological level in human patients and animal models with pathogenic mutations in *AFG3L2* or *SPG7* (Casari et al, 1998; Nolden et al, 2005; Almajan et al, 2012). Our data also establish the hierarchy to these molecular events, arising from the progressive failure in the quality control of mitochondrial nascent chain synthesis. Our findings did not identify the specific type of mitochondrial nascent chain proteotoxicity that occurs with AFG3L2 dysfunction. However, we provided experimental evidence by which the translation of a specific class of aberrant mitochondrial mRNA also activates the same stress response pathway. Thus, the proteotoxic trigger with AFG3L2 dysfunction may arise during translation of aberrant mitochondrial mRNAs. Currently, we have little understanding on the frequency by which mistakes are generated during the nucleolytic processing of the mitochondrial polycistronic RNA messages. Presumably, these errors in gene expression are resolved quickly by RNA surveillance mechanisms (Keiler, 2015) so as not to activate an organelle stress response. To test this attractive hypothesis requires experimentation beyond the scope of the present study but promises to be an exciting avenue of future research.

Mitochondrial dysfunction generates a heterogeneous collection of clinical disorders that cannot be explained simply by a defect in oxidative phosphorylation (Suomalainen & Battersby, 2018). Understanding the specific molecular defects and ensuing stress responses is likely key to the pathogenesis. To this end, it is perhaps not surprising that we found only a single class of additional pathogenic mutation (*MT-ATP6* m.9205delTA) that activated the mitochondrial translation stress response, but only following the stress of heat shock. Acute febrile infections are a critical event in the natural history of mitochondrial disorders (Thorburn et al, 2003; Magner et al, 2015). It could be argued the heat stress used in our experiments is high, but it is important to point out that we used cultured human skin fibroblasts, not neurons, which would be exposed to such temperatures under physiological conditions across the globe. Interestingly, the chaperone function of the AAA domain in the AFG3L2 complex, and not the proteolytic capacity, was critical for resolving the stress response initiated by nascent chain synthesis of the *MT-ATP6* m.9205delTA during heat shock. Other mitochondrial AAA protease complexes, such as CLPXP, also have a distinct chaperone role independently of proteolytic function (Kardon et al, 2015). Collectively, the findings of this study demonstrated how modulating the rate of mitochondrial protein synthesis in a context-specific case can be beneficial to organelle homeostasis and cell fitness. Such an approach may be an avenue to develop for patients with AFG3L2 and SPG7 mutations.

Numerous studies have established OPA1 as a central factor for the dynamics and ultrastructure of mitochondrial membranes with

---

**Figure 5.   AFG3L2 dysfunction remodels mitochondrial inner membrane ultrastructure.**
**(A)** Representative transmission electron micrographs of human fibroblasts with the indicated *MT-ATP6* genotypes treated with siRNAs. SC, scrambled sequence. **(B)** Mitochondrial inner membrane morphology by electron tomographic models from three perspectives in wild-type control human fibroblasts treated with the indicated siRNAs. Numbers indicate the position of cristae junctions along a single crista invagination in different orientations. **(C)** Same as (B) but in human fibroblasts with the indicated *MT-ATP6* genotype. IMS, intermembrane space.

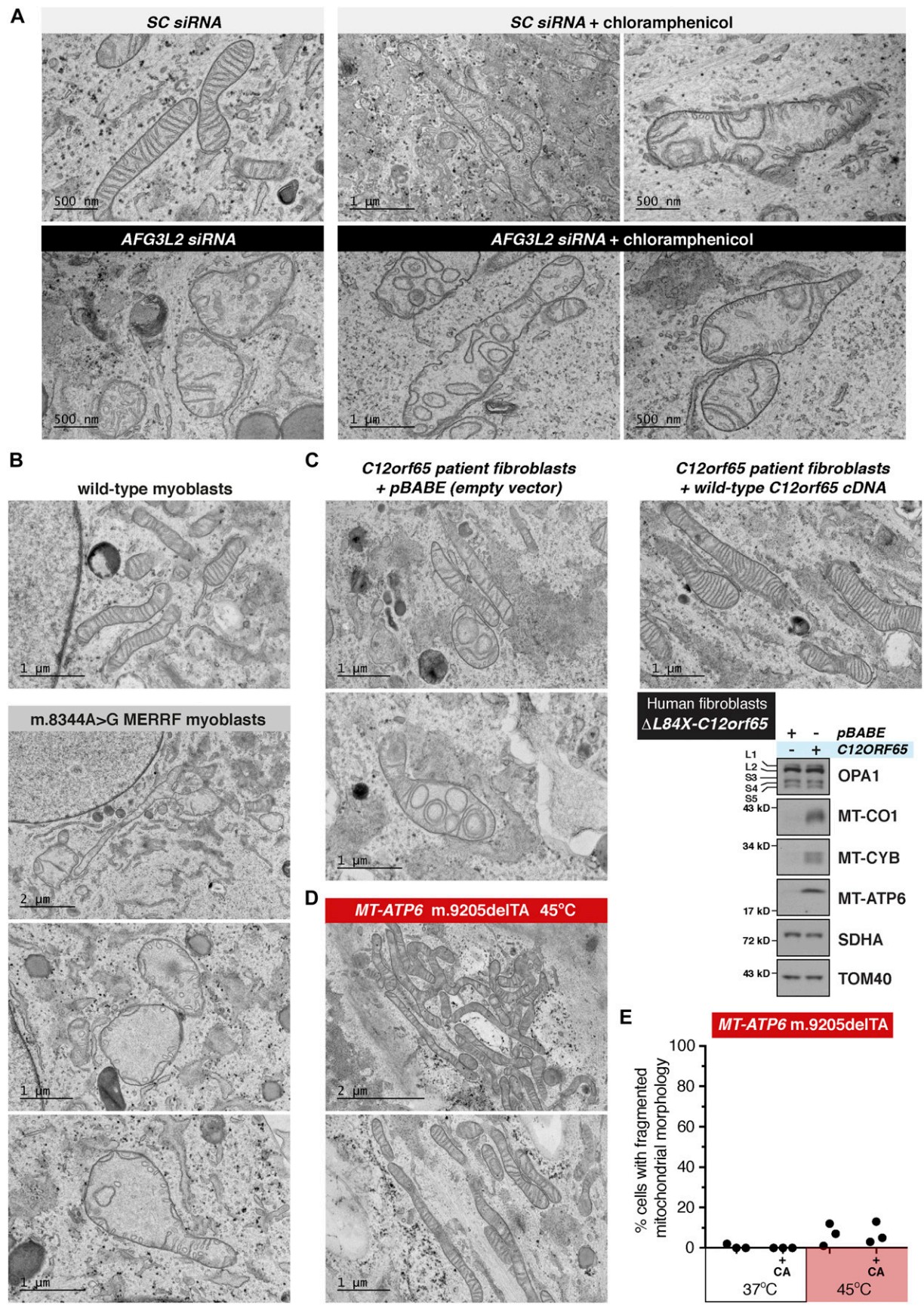

**A**

SC siRNA
500 nm

SC siRNA + chloramphenicol
1 µm
500 nm

AFG3L2 siRNA
500 nm

AFG3L2 siRNA + chloramphenicol
1 µm
500 nm

**B**

wild-type myoblasts
1 µm

m.8344A>G MERRF myoblasts
2 µm
1 µm
1 µm

**C**

C12orf65 patient fibroblasts
+ pBABE (empty vector)
1 µm
1 µm

C12orf65 patient fibroblasts
+ wild-type C12orf65 cDNA
1 µm

Human fibroblasts
ΔL84X-C12orf65

| | + | – | pBABE |
|---|---|---|---|
| | – | + | C12ORF65 |

L1
L2
S3
S4
S5 — OPA1
43 kD — MT-CO1
34 kD — MT-CYB
MT-ATP6
17 kD
72 kD — SDHA
43 kD — TOM40

**D**

MT-ATP6 m.9205delTA 45°C
2 µm
1 µm

**E**

MT-ATP6 m.9205delTA

% cells with fragmented mitochondrial morphology

100
80
60
40
20
0

+ CA      + CA
37°C        45°C

the proteolytic processing of the membrane-tethered form proposed as a key regulator step (Friedman & Nunnari, 2014; MacVicar & Langer, 2016). An outstanding question, however, remains on what is a direct function of OPA1 in cristae morphogenesis from an indirect or downstream response. Because OPA1 proteolytic processing is associated with pleiotropic phenotypes, it is important to resolve this distinction. Our data revealed dynamic cross-talk between OPA1 and mitochondrial protein synthesis, suggesting that many functions attributed to the stress-induced proteolytic processing of OPA1 could also be due to altered protein synthesis by mitochondrial ribosomes. Moreover, our electron tomograms of mitochondria in situ demonstrated OPA1 proteolytic processing is not accompanied with alterations to the cristae junctions or acute changes to the inner membrane ultrastructure. This is in contrast to observations from experiments with isolated mitochondria using the proapoptotic inducer BID (Frezza et al, 2006). Therefore, the long membrane-tethered OPA1 does not appear to play a strict role in cristae junction formation in all cases, suggesting the general applicability of this model should be reconsidered and may only be relevant to dying cells or isolated mitochondria.

Currently, the proximity of OPA1 and OMA1 to the sites of mitochondrial protein synthesis is not known, but the apparent tight coupling between these systems suggests a physical juxtaposition. Severing the OPA1 membrane tether appeared to correlate with a loss of mitochondrial gene expression in many cases. However, when the AFG3L2 E575Q mutant was overexpressed in the *MT-ATP6* m.9205delTA background, we blocked OMA1 activation and ribosome decay but not the attenuation in mitochondrial protein synthesis associated with heat shock, suggesting that the OPA1 stress-induced proteolytic processing is not a regulatory switch for mitochondrial gene expression. Thus, the physiological rationale for this regulatory event remains unknown. The bulk of the investigations into the regulation of OMA1 activation have used ectopic addition of agents such as CCCP (Ishihara et al, 2006; Ehses et al, 2009; Head et al, 2009; Baker et al, 2014); however, these compounds are severely toxic to cells, producing a cellular state of a magnitude neither physiological nor relevant to human disease. The use of these compounds will not reveal the endogenous signals that trigger this key regulatory event in mitochondrial homeostasis. Future work should seek to determine the functional interplay between OPA1 and mitochondrial gene expression because our data robustly demonstrated failures in the quality control of nascent chain synthesis is a major OMA1 trigger.

Activation of the mitochondrial translation stress response induced mitochondrial ribosome decay. Previously, we showed how protein over-accumulation and dissipation of the membrane potential also induced mitochondrial rRNA and mRNA decay (Richter et al, 2013, 2015) but occurred on a slower time scale than we observed in the *MT-ATP6* m.9205delTA patient fibroblasts with heat shock. Mammalian mitochondrial ribosomes are composed of more than 80 individual proteins, two rRNAs, and a tRNA (Amunts et al, 2015; Greber et al, 2015) and require a plethora of factors to coordinate assembly many of which are now being discovered (Ott et al, 2016; Brown et al, 2017). Our data provide an example of organelle dysfunction that triggers a degradation process specific to mitochondrial ribosomes and show the relevance of this decay as a molecular mechanism in the pathogenesis of human disease. These matrix localized degradation pathway(s) will be independent of processes that involve large-scale organelle turnover, such as mitophagy, because we did not observe any changes to mitochondrial DNA copy number or core import machinery (e.g., TOM40) that are degraded in Parkin-mediated mitophagy (Yoshii et al, 2011). In the cytosol, ribosome degradation is mediated by RNA surveillance pathways following severe aberrations in translation elongation (Lykke-Andersen & Bennett, 2014). The mechanisms for these processes in mitochondria are largely unknown but open up an exciting avenue for future research.

# Materials and Methods

## Cell culture

Human and mouse fibroblasts, and cell lines were cultured at 37°C and 5% $CO_2$ in Dulbecco's modified Eagle's medium (Lonza) with high glucose supplemented with 10% fetal bovine serum, 1× glutamax, and 50 μg/ml uridine. All primary cells were immortalized by expressing E7 and hTERT (Lochmüller et al, 1999) unless previously immortalized by the donating laboratory. All cells tested negative for mycoplasma infection (PromoKine). Human myoblast cultures (kind gift from Eric Shoubridge) were grown in myoblast medium from Lonza. Chloramphenicol (Sigma-Aldrich) was used at a dose of 400 μg/ml. Heat shock was administered by adding pre-warmed media to the plate and immediate transfer to an incubator set at 45°C. In the case of chloramphenicol treatment, the cells were treated for 72 h before and during heat shock. Stealth siRNAs (Thermo Fisher Scientific: AFG3L2, HSS116886, 5'-GAGUAGUAGACAGAUUGGAA-GUCGU; Atp5b, MSS202262, 5'-GAUCACCUCGGUGCAGGCUAUCUAU; PHB2-1, HSS117606, 5'-ACGAUCGCCACAUCACAGAAUCGUA; PHB2-2, HSS117604, 5'-CCAAAGACCUACAGAUGGUGAAUAU; STOML2, HSS121274, 5'-GCCUCCGUUAUGAGAUCAAGGAUAU; TMEM70, HSS123635, 5'-CGAGUCUGAUUGGCCUUACAUUUCU; and scrambled control, 12935300) were transfected on day 1 and 3 with Lipofectamine RNAiMAX (Thermo Fisher Scientific) and collected on day 8 for analysis unless indicated otherwise. All siRNA knockdowns were evaluated by immunoblotting using specific antibodies. Retrovirus was generated by transient transfection of retroviral plasmids into

---

**Figure 6. Global defects to mitochondrial protein synthesis remodels cristae ultrastructure.**
**(A)** Representative transmission electron micrographs of human fibroblasts treated with the indicated siRNAs with or without chloramphenicol (CA). **(B)** Representative transmission electron micrographs of human myoblasts with the same nuclear genetic background that are either homoplasmic for wild-type mitochondrial DNA (mtDNA) or the m.8344A>G tRNA[Lys] mutation. **(C)** Representative transmission electron micrographs of human fibroblasts from a patient with a pathogenic mutation in *C12orf65* stably transduced with an empty vector or the wild-type *C12orf65* cDNA. Bottom right, immunoblotting of whole-cell lysates with the indicated antibodies. **(D)** Representative transmission electron micrographs of human fibroblasts with the *MT-ATP6* m.9205delTA mutation heat shocked for 4 h at 45°C. **(E)** Scatter plot quantification of mitochondrial morphology under light microscopy in human fibroblasts with the *MT-ATP6* m.9205delTA mutation grown at 37°C or heat shocked for 4 h at 45°C with or without CA. Data are from three independent experiments. SC, scrambled sequence.

the Phoenix amphotropic packaging line. The cells were directly in experiments following selection with puromycin or blasticidin.

## Cloning

Wild-type cDNAs (AFG3L2, clone BC065016; COX10, clone BC000060) were obtained from the ORFeome or Mammalian Genome Collection. These cDNAs were in a Gateway entry vector (pENTR221 or pCMV-SPORT6) or cloned into pDONR201 using PCR with KAPA HiFi followed by recombination. All full-length cDNAs were recombined into Gateway-converted pBABE-puro and pMXs-IRES-Blasticidin with LR Clonase II (Thermo Fisher Scientific). All cDNAs were thoroughly sequenced by Sanger sequencing for verification initially and after all PCR manipulations. Site-directed mutagenesis of E575Q in AFG3L2 was performed using Agilent primer design with PCR (KAPA HiFi), purification, DpnI (NEB) digest, and transformation into XL1 Blue chemically competent cells. Sanger sequencing was used to validate mutagenesis. Mutant cDNA was recombined into Gateway-converted pBABE-puro and pMXs-IRES-Blasticidin with LR Clonase II.

## Immunoblotting

Cells were solubilized in phosphate-buffered saline, 1% dodecyl-maltoside (DDM), 1 mM PMSF, and complete protease inhibitor (Thermo Fisher Scientific). Protein concentrations were measured by the Bradford assay (Bio-Rad). Equal amounts of proteins were separated by Tris-glycine SDS–PAGE and transferred to nitrocellulose by semi-dry transfer or by Tricine-SDS–PAGE onto polyvinylidene difluoride for proteins smaller than 20 kD (Schägger, 2006). Membranes were blocked in Tris-buffered saline, 0.1% Tween 20 (TBST) with 1% milk at room temperature for 1 h and then primary antibodies were incubated overnight at +4°C in 5% BSA/TBST and detected the following day with secondary HRP conjugates (Jackson ImmunoResearch) using ECL with film. Primary antibodies from Proteintech Group: AFG3L2 (14631-1-AP, 1:5,000); ATP5B (17247-1-AP, 1:5,000); uL11m (15543-1-AP, 1:20,000); bS18b (16139-1-AP, 1:2,000); mS26 (15989-1-AP, 1:5,000); mS27 (17280-1-AP, 1:5,000); MT-ATP6 (55313-1-AP, 1:2,000); MT-CYB (55090-1-AP, 1:1,000); PHB2 (66424-1-Ig, 1:5,000); STOML2 (10348-1-AP, 1:5,000); and TMEM70 (20388-1-AP, 1:5,000). Primary antibodies from Abcam/Mitosciences: MT-CO1 (1D6E1A8, 1:500) and SDHA (C2061/ab14715, 1:10,000), and primary antibody from Santa Cruz: TOM40 (sc-11414, 1:5,000). Representative data of independent experiments were cropped in Adobe Photoshop with only linear corrections to brightness applied.

## TMRM staining

HEK293 cells were treated with siRNAs against AFG3L2 and scrambled sequence then stained with 200 nM TMRM (Life Technologies). For a positive control, the cells were treated with CCCP (10 $\mu$M). After TMRM staining, the cells were analyzed by flow cytometry (Accuri C6, Becton Dickinson or a BD LSR II). Independent experiments were performed and quantified. All siRNA experiments were subsequently validated by immunoblotting.

## Metabolic labelling of mitochondrial protein synthesis

Mitochondrial protein synthesis was analyzed by metabolic labelling with $^{35}$S methionine/cysteine (Richter et al, 2015). The cells were pretreated with anisomycin (100 $\mu$g/ml) to inhibit cytoplasmic translation then pulsed with 200 $\mu$Ci/ml $^{35}$S Met–Cys (EasyTag-Perkin Elmer). In chase experiments, the cells were pulsed for 30 min with radiolabel then the medium removed and replaced with fresh medium lacking the radioisotope for the indicated time. Equal amounts of sample protein were first treated with Benzonase (Thermo Fisher Scientific) on ice and then resuspended in 1× translation loading buffer (186 mM Tris-Cl, pH 6.7, 15% glycerol, 2% SDS, 0.5 mg/ml bromophenol blue, and 6% $\beta$-mercaptoethanol). 12%–20% gradient Tris-glycine SDS–PAGE gels were used to separate the samples and then dried for exposure with a phosphoscreen and scanned with a Typhoon 9400 or Typhoon FLA 7000 (GE Healthcare) for quantification. The gels were rehydrated in water and Coomassie stained to confirm loading.

## Mass spectrometry analysis

Mitochondria were isolated (QIAGEN) from control and m.9205delTA MT-ATP6 fibroblasts cultured at 37°C or following a 4-h heat shock at 45°C. Mitochondria were lysed (PBS, 1% DDM, 1 mM PMSF, and complete protease inhibitor [Thermo Fisher Scientific]) on ice for 20 min and then centrifuged at 20,000 $g$ for 20 min at +4°C. Insoluble materials were discarded and proteins in the supernatant were precipitated with acetone.

The protein pellet was dissolved in ProteaseMax (Promega) in 50 mM $NH_4HCO_3$, and the proteins were reduced, alkylated, and in-solution digested with trypsin (Promega) according to the manufacturer's instructions. Peptides were desalted and concentrated before mass spectrometry by the STAGE-TIP method using a C18 resin disk (3M Empore). The peptides were eluted twice with 0.1% formic acid/50% ACN, dried, and solubilized in 7 $\mu$l 0.1% formic acid for mass spectrometry analysis. Each peptide mixture was analyzed on an Easy nLC1000 nano-LC system connected to a quadrupole Orbitrap mass spectrometer (QExactivePlus; Thermo Electron) equipped with a nanoelectrospray ion source (EasySpray/Thermo Fisher Scientific). For the liquid chromatography separation of the peptides, we used an EasySpray column capillary of 25 cm bed length (C18, 2 $\mu$m beads, 100 Å, 75 $\mu$m inner diameter, Thermo). The flow rate was 300 nl/min, and the peptides were eluted with a 2%–30% gradient of solvent B in 60 min. Solvent A was aqueous 0.1% formic acid and solvent B 100% acetonitrile/0.1% formic acid. The data-dependent acquisition automatically switched between MS and MS/MS mode. Survey full scan MS spectra were acquired from a mass-to-charge ratio (m/z) of 400 to 1,200 with the resolution R = 70,000 at m/z 200 after accumulation to a target of 3,000,000 ions in the quadruple. For MS/MS, the 10 most abundant multiple-charged ions were selected for fragmentation on the high-energy collision dissociation cell at a target value of 100,000 charges or maximum acquisition time of 100 ms. The MS/MS scans were collected at a resolution of 17,500. Target ions already selected for MS/MS were dynamically excluded for 30 s.

The resulting MS raw files were submitted to the MaxQuant software version 1.6.1.0 for protein identification using the

Andromeda search engine. The UniProt human database (October 2017) was used for database searches. Carbamidomethyl (C) was set as a fixed modification, and protein N-acetylation and methionine oxidation were set as variable modifications. First search peptide tolerance of 20 ppm and main search error 4.5 ppm were used. Trypsin without proline restriction enzyme option was used, with two allowed miscleavages. The minimal unique + razor peptides number was set to 1, and the allowed false discovery rate was 0.01 (1%) for peptide and protein identification. Label-free quantitation () was used with default settings. Known contaminants as provided by MaxQuant and identified in the samples were excluded from further analysis. Statistical analysis was performed with Perseus software version 1.5.6.0. The label-free quantitation data were log10 transformed, filtered to include only proteins, which were quantified in at least in three of five replicates in at least one group, and missing values were imputed with values representing a normal distribution with default settings in Perseus. To find statistically significant differences between the groups, $t$ test was performed using permutation-based false discovery rate with 0.05 cut off (Tyanova et al, 2016a, 2016b). The MS data are available in the PRIDE database under accession number PXD012416.

## Southern blotting

Total DNA was isolated from cultured fibroblasts as described in QIAamp DNA Mini Kit (QIAGEN). DNA (5 µg) was digested with PvuII, then separated on a 0.8% agarose gel followed by alkaline transfer to Hybond-XL membrane (GE Healthcare). Two oligonucleotide probes for mtDNA (5′-GGCTCCAGGGTGGGAGTAGTTCCCTGC; 5′-CCTCCCGAATCAACCCTGACCCCTCTCC) and three for 18S rDNA (5′-GGCCCGAGGTTATCTAGAGTCACC; 5′-TATTCCTAGCTGCGGTATCCAGGC; and 5′-ACCATCCAATCGGTAGTAGCGACG) were 5′ end labelled with $\gamma$ $^{32}$P ATP by T4 polynucleotide kinase (NEB) and hybridized (50% formamide, 7% SDS, 0.25M sodium phosphate, pH 7.2, 10 mM EDTA, pH 8.0, 0.24M NaCl$_2$) overnight at 37°C. Membranes were washed once with 2× SSC/0.1%SDS for 60 min, followed by 0.5× SSC/0.1% SDS for 60 min, and finally in 0.1× SSC/0.1% SDS for 30 min. All washings were performed at 37°C. The membranes were dried and then exposed to a Phosphoscreen (GE Healthcare) and scanned with a Typhoon 9400 (GE Healthcare).

## Northern blotting

Total cellular RNA was isolated using Trizol (Invitrogen) according to the manufacturer's instructions. For all samples, 5 µg of total RNA was run through 1.2% agarose-formaldehyde gels and transferred to Hybond-N$^+$ membrane (GE Healthcare) by neutral transfer. T4 polynucleotide kinase (NEB) 5′-radiolabelled oligonucleotides were used for detection of mitochondrial transcripts. Oligonucleotides: 12S, 5′-GTTAATCGTGTGACCGCGGTGGCTGGC; 16S, 5′-GCTGTGTTATGCCCGCCTCTTCACGGG; and 18S, 5′-ACCATCCAATCGG-TAGTAGCGACG. Hybridization (25% formamide, 7% SDS, 1% BSA, 0.25M sodium phosphate, pH 7.2, 1 mM EDTA, pH 8.0, and 0.25M NaCl$_2$) was performed for 16–20 h at 37°C. Membranes were washed (2× SSC/0.1% SDS) and then dried for exposure on a phosphoscreen (GE Healthcare) and scanned with a Typhoon 9400 (GE Healthcare).

## Amplicon generation and Illumina sequencing

Genomic DNA was extracted from patient cells known to contain m.9205.delTA and wild-type control cells (Flp-In T-REx 293; Thermo Fisher Scientific) using NucleoSpin Tissue kit (Macherey-Nagel GmbH & Co. KG) according to the manufacturer's instructions. Three PCR amplicons spanning the deletion site were designed (amplicon 1 [3.90 kb] [primers 5′-tgtaaaacgacggccagtCGCAAGTAGGTCTACAAGACG; 5′-caggaaacagctatgaccATAGAGAGGTAGAGTTTTTTTCGTG]; amplicon 2 [3.16 kb] [primers 5′-tgtaaaacgacggccagtATTCTTATCCTACCAGGCTTCG; 5′-caggaaacagctatgaccAATGTTGAGCCGTAGATGCC]; and amplicon 3 [2.49 kb] [primers 5′-tgtaaaacgacggccagtCCCCATACTAGT-TATTATCGAAACC; 5′-caggaaacagctatgaccGGCTTCGACATGGGCTTT]) and amplified in 20-µl reactions with Phusion High-Fidelity DNA Polymerase (New England Biolabs). Cycling conditions were as follows: 95°C for 1 min, 35 cycles of 95°C for 30 s, 60°C for 30 s, and 72°C for 2 min followed by 72°C for 3 min. The PCR products were purified using NucleoSpin Gel and PCR clean-up kit (Macherey-Nagel GmbH & Co. KG) according to the manufacturer's instructions including second wash step and elution twice to 15 µl of elution buffer. Library preparations and Illumina sequencing reactions were performed by the Max Planck-Genome-Centre Cologne, Germany (http://mpgc.mpipz.mpg.de/home/).

Sequencing reads were trimmed with Flexbar v2.5 (Dodt et al, 2012) for TruSeq adapters and base call quality (default parameters except -q 28 -m 50 -ae ANY -ao 10). Reads containing the M13 tag sequences in forward or reverse orientation were removed by Flexbar (parameters -bo 15 -bt 2 -bu -be ANY) to avoid primer-containing reads and highly biased coverage at the amplicon edges. Unassigned reads (without the M13 tag) were aligned with BWA v0.7.12-r1039 invoking mem (Li & Durbin, 2010; Li, 2013) (parameters -T 19 -B 4 -L 5,4) to the human mitochondrial reference sequence (NC_012920.1). Uniquely aligned reads (SAMtools view -q 1) were converted to bam format, sorted, and indexed with SAMtools (Li et al, 2009). Per base coverage was determined with bedtools v2.22.1 (Quinlan & Hall, 2010) genomecov (parameters -split -d). Variants were detected with Lofreq* v2.1.2 (Wilm et al, 2012) as follows. First, the indel qualities were set by command lofreq indelqual –dindel, and then variants were called including indels with minimum base call quality 30 by command lofreq call-parallel –pp-threads 20 -N -B -q 30 -Q 30 –call-indels –no-default-filter. Variant results were further filtered for minimum variant quality Phred score 70 and maximum strand bias Phred score 60 (strand bias filter applied only if ≥85% of the variant supporting reads were on a single strand) by command lofreq filter –no-defaults –snvqual-thresh 70 –sb-incl-indels -B 60. Next, the variants were filtered for minimum of 15 variant supporting reads using snpSift filter with the expression DP*AF ≥ 15 and for minimum of three variant supporting reads on each strand (expression DP4[2] ≥ 3 & DP4 [3] ≥ 3). Finally, only variants and indels present at positions covered by minimum 100,000 reads and minimum variant allele frequency (AF value) of 1% were accepted. To understand the level of the reference genome bias, the aforementioned data analysis steps were repeated using a reference genome containing the deletion (9206.delA and 9205.delT). The detected indel frequencies were then compared.

## Isokinetic sucrose gradients

Medium from cultured cells grown on a 150-mm plate was rapidly removed, and the cells were flash-frozen on dry ice. The plates were immediately transferred to wet ice to thaw. The cells were washed two times with ice-cold PBS, lysed on the plate (50 mM Tris, pH 7.2, 10 mM Mg(Ac)$_2$, 40 mM NH$_4$Cl 100 mM KCl, 1% DDM, 1 mM ATP, 400 $\mu$g/ml chloramphenicol, and 1 mM PMSF), and transferred to a 1.5-ml centrifuge tube for a 20-min incubation on ice. The samples were centrifuged for 10 min at 20,000 $g$ at 4°C. The supernatant was loaded on top of a 16-ml linear 10%–30% sucrose gradient (50 mM Tris, pH 7.2, 10 mM Mg(Ac)$_2$, 40 mM NH$_4$Cl 100 mM KCl, 1 mM ATP, 400 $\mu$g/ml chloramphenicol, and 1 mM PMSF) and centrifuged for 15 h at 4°C and 74,400 $g$ (Beckman SW 32.1 Ti). 24 equal-volume fractions were collected from the top for either trichloroacetic acid precipitation for protein immunoblotting or for RNA isolation.

## Light microscopy

Fibroblasts were grown on coverslips and washed several times in PBS, fixed in 4% paraformaldehyde for 15 min, washed in PBS, then treated with 100% methanol for 5 min, and washed again in PBS. The cells were blocked in 5% BSA/PBST (PBS + 0.1% Tween 20) and then incubated with rabbit anti-TOM20 (sc-11414, 1:250; Santa Cruz) for 1 h at room temperature or overnight +4°C. The cells were washed several times in PBST before incubation with anti-rabbit Alexa 488 1:1,000 (Life Technologies) for 1 h and then washed several times in PBST. Coverslips were mounted with DABCO/MOWIOL on glass slides for imaging with a Zeiss inverted microscope at room temperature using a Plan-Apochromat 63×/1.40 Oil and slider assembly Apo-Tome2. Images were exported into ImageJ using the Fiji plug-in to apply brightness and contrast adjustments.

## Transmission electron microscopy

Cultured cells were grown on glass coverslips and then fixed with 2% glutaraldehyde in Na-phosphate buffer, pH 7.4, for 30 min at room temperature. After fixing, the samples were washed 3× 2 min with Na-phosphate buffer, pH 7.4. The samples were then post-fixed with 1% osmium tetroxide for 1 h at room temperature, dehydrated with a graded series of ethanol (70%, 96%, and 100%), incubated with transitional solvent acetone, and finally embedded gradually in Epon (TAAB). Images were acquired on a Jeol JEM-1400 transmission electron microscope with a Gatan Orius 1000B bottom-mounted CCD camera.

## Electron tomography and modelling

Samples were prepared as described for TEM. Dual axis tilt series (Mastronarde, 1997) were recorded from one or consecutive semi-thick (250 nm) sections using Tecnai FEG 20 (FEI Company) microscope operating at 200 kV. The sections were tilted at one-degree intervals using a high-tilt specimen holder (model 2020; E.A. Fischione Instruments) between ±62°. Images were acquired with SerialEM software (Mastronarde, 2005) using a 4 k × 4 k Ultrascan 4000 CCD camera (Gatan Corporation) at nominal magnification of 9,600× or 11,500×. The alignment of the tilt series and

reconstructions were done with IMOD software package (Kremer et al, 1996) using 10-nm colloidal gold particles as fiducial markers. Tomographic reconstructions were segmented with MIB software (Belevich et al, 2016) and visualization was performed using Amira (VSG, FEI Company).

## High-resolution respirometry

Oxygen consumption rates were determined by substrate–uncoupler–inhibitor titration protocols in permeabilized human fibroblasts treated with siRNA using a high-resolution oxygraph (OROBOROS instrument). Values of specific oxygen consumption rates determined as pmol/(s*millions of cells) and are given as ratios of different respiration states. Briefly, ~2 × 10$^6$ cells were resuspended in MiR05 (0.5 mM EGTA, 3 mM MgCl$_2$, 60 mM lactobionic acid, 20 mM taurine, 10 mM KH$_2$PO$_4$, 20 mM Hepes, 110 mM D-sucrose, and 1% fat-free BSA) and used per measurement. Oxygen consumption rates were measured in digitonin-permeabilized (1 $\mu$l of 8 mM per 1 × 10$^6$ cells) in the presence of pyruvate–glutamate–malate (state 4 respiration on complex I), after addition of ADP (state 3 respiration on complex I), succinate (state 3 respiration on complex I and II), FCCP titration (maximal uncoupled respiration), and finally ascorbate with TMPD followed by azide (complex IV activity).

# Supplementary Information

# Acknowledgements

We thank LS Churchman; E Dufour, C Dunn, K Holmström, H Jacobs, T Langer, V Paavilainen, and E Rugarli for valuable discussion; EA Shoubridge, T Langer, and CL Otín for generously sharing reagents; and the Max Planck-Genome-Centre Cologne, the University of Helsinki Electron Microscopy Unit and Genome Biology Unit. This work was supported by the Academy of Finland Centre of Excellence on Mitochondria to BJ Battersby; an Academy of Finland postdoctoral award (to U Richter); the Ella and Georg Ehrnrooth Foundation (to F Suomi). RW Taylor is supported by the Wellcome Centre for Mitochondrial Research (203105/Z/16/Z), the Medical Research Council Centre for Translational Research in Neuromuscular Disease, Mitochondrial Disease Patient Cohort (UK) (G0800674), the Lily Foundation, and the UK NHS Highly Specialised Service for Rare Mitochondrial Disorders of Adults and Children.

## Author Contributions

U Richter: conceptualization, formal analysis, and investigation.
KY Ng: conceptualization, formal analysis, and investigation.
F Suomi: conceptualization, formal analysis, and investigation.
P Marttinen: formal analysis and investigation.
T Turunen: formal analysis and investigation.
C Jackson: formal analysis and investigation.
A Suomalainen-Wartiovaara: resources.
H Vihinen: formal analysis and methodology.
E Jokitalo: formal analysis and methodology.

TA Nyman: formal analysis and investigation.
MA Isokallio: formal analysis and investigation.
JB Stewart: formal analysis and investigation.
C Mancini: resources.
A Brusco: resources.
S Seneca: resources.
A Lombès: resources.
RW Taylor: resources.
BJ Battersby: conceptualization, formal analysis, supervision, funding acquisition, investigation, project administration, and writing—original draft, review, and editing.

## Conflict of Interest Statement

The authors declare that they have no conflict of interest.

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
