## [Reviewer comments · Life Science Alliance]

Life Science Alliance

A MITOCHONDRIAL STRESS RESPONSE TRIGGERED BY DEFECTS IN PROTEIN SYNTHESIS QUALITY CONTROL

Uwe Richter, Kah Ng, Fumi Suomi, Paula Marttinen, Taina Turunen, Christopher Jackson, Anu Suomalainen-Wartiovaara, helena Vihinen, Eija Jokitalo, Tuula Nyman, Marita Isokallio, James Stewart, Cecilia Mancini, Alfredo Brusco, Sara Seneca, Anne Lombès, Robert Taylor, and Brendan Battersby

DOI: <https://doi.org/10.26508/lisa.201800219>

Corresponding author(s): Brendan Battersby, University of Helsinki

Review Timeline:

Submission Date:	2018-10-20
Editorial Decision:	2018-11-08
Revision Received:	2018-11-30
Editorial Decision:	2018-12-08
Revision Received:	2018-12-21
Editorial Decision:	2019-01-14
Revision Received:	2019-01-16
Accepted:	2019-01-17

Scientific Editor: Andrea Leibfried

Transaction Report:

November 8, 2018

Re: Life Science Alliance manuscript #LSA-2018-00219-T

Dr. Brendan J Battersby
University of Helsinki
Institute of Biotechnology
Viikinkaari 5D
Helsinki, Uusimaa 790
Finland

Dear Dr. Battersby,

Thank you for submitting your manuscript entitled "A MULTI-LAYERED STRESS RESPONSE TRIGGERED BY DEFECTS IN MITOCHONDRIAL PROTEIN SYNTHESIS QUALITY CONTROL" to Life Science Alliance. The manuscript was assessed by expert reviewers, whose comments are appended to this letter.

As you will see, all reviewers appreciate your analyses. However, they also think that it remains unclear why aberrant nascent chains are so toxic in mitochondria and how AFG3L2 levels affect ribosomes. Importantly, they also point out that the manuscript is very difficult to read and not easily accessible to readers. Finally, additional controls and assays are needed to better support your conclusions.

We would like to invite you to revise your work for publication in Life Science Alliance following the reviewers' constructive input. We think it would be most productive to discuss the revision further with you and based on a preliminary point-by-point response, to see whether you have data at hand that could (partially) address the concern regarding lack of sufficient insight and to hear from you how you would address the other comments made (re-writing (rev#1), specific points of rev#2, points 1-4 of rev#3). Please send such response by email to us so that we can further discuss.

When submitting the revision, please include a letter addressing the reviewers' comments point by

point.

Thank you for this interesting contribution to Life Science Alliance. We are looking forward to receiving your revised manuscript.

Sincerely,

- A letter addressing the reviewers' comments point by point.
- An editable version of the final text (.DOC or .DOCX) is needed for copyediting (no PDFs).
- High-resolution figure, supplementary figure and video files uploaded as individual files: See our detailed guidelines for preparing your production-ready images, <http://life-science-alliance.org/authorguide>
- Summary blurb (enter in submission system): A short text summarizing in a single sentence the study (max. 200 characters including spaces). This text is used in conjunction with the titles of papers, hence should be informative and complementary to the title and running title. It should describe the context and significance of the findings for a general readership; it should be written in the present tense and refer to the work in the third person. Author names should not be mentioned.

B. MANUSCRIPT ORGANIZATION AND FORMATTING:

Full guidelines are available on our Instructions for Authors page, <http://life-science-alliance.org/authorguide>

Reviewer #1 (Comments to the Authors (Required)):

Richter and colleagues examine processing of the mitochondrial protease OPA1 as an indicator of a proteotoxic stress response mediated by OMA1. Most interestingly, they convincingly demonstrate that an array of mitochondrial stressors can be alleviated by inhibiting protein synthesis from mitochondrial ribosomes. This is potentially quite interesting. Unfortunately, the authors also claim they understand the loss of mitochondrial ribosomes and the remodeling of the mitochondrial inner membrane that occurs in human disease. This is not supported.

A major problem with the manuscript is the writing/presentation of the Results. The Introduction and Discussion are very nicely written and accessible to a broad audience. The results section is NOT. It is very dense and nearly impossible to follow.

For example, the following is the title of one of the sections:
"Mitochondrial protein synthesis required for lamellar cristae"
This is not surprising.

Reviewer #2 (Comments to the Authors (Required)):

AFG3L2 forms a matrix facing AAA protease in mitochondria. In this study, Richter et al. analyzed the functional consequences of the defects in AFG3L2 in mitochondria. They carefully followed the effects of AFG3L2 knock down to reveal that translation-dependent activation of another protease OMA1 took place first, then loss of mitochondrial ribosomes, and OMA1-mediated processing of the dynamin-related GTPase OPA1 followed, and finally alteration in the inner membrane ultrastructure was seen. The observations are interesting, and raise many questions to stimulate further studies. However, the results at the moment remain more or less descriptive, and lack insight into the mechanism underlying the observed phenomena. In particular, it is not clear what caused translation-dependent toxicity when AFG3L2 was knocked down. Why degradation of mitochondrially translated proteins by AFG3L2 is required even under normal physiological conditions? To address this point, the authors could monitor the fate of translation products of mitochondrially encoded proteins, probably by RI labeling (under the condition where cytosolic protein synthesis is inhibited), with and without AFG3L2 knockdown. It is not clear, either, why mitochondrial ribosome levels decreased when AFG3L2 was knocked down; clearly activation of Oma1 facing the intermembrane space is not directly involved in this reduction of mitochondrial ribosome proteins in the matrix. Without mechanistic insight into these processes, the present study stays too premature.

Specific points

Fig. 1G - The asterisk indicates a wrong band for AFG3L2?

Fig. S1A - The authors observed that prohibitin knock down caused Opa1 processing, by OMA1, but is this related to the AFG3L2-dependent OMA1 activation?

Fig. 2F - A control experiment to show inhibition of cytosolic translation by anisomycin is missing.

Reviewer #3 (Comments to the Authors (Required)):

The assembly of heterooligomeric protein complexes depends on the coordinated expression of its subunits and surveillance by proteolytic systems. This is particularly complicated in the case of the respiratory chain and mitochondrial ribosomes which consist of components of dual genetic origin. The mechanisms by which the cytosolic ubiquitin system controls synthesis, folding and assembly of cytosolic proteins is well understood. However, we know only little about the balance of synthesis, assembly and degradation of mitochondrially encoded proteins, and most of what we know is based on observations in yeast.

In the present study, Richter et al. elucidate the interplay of mitochondrial proteases (Oma1, AFG3L2), the mitochondrial ribosome and the mitochondrial translation products. They show that the translation of an aberrant mitochondrial mRNA induces a specific membrane stress response characterized by OMA1 activation and reduced levels of mitochondrial ribosomal proteins. Moreover, the authors propose that the AFG3L2 AAA complex facilitates the unfolding of nascent mitochondrial translation products, which are of profound proteotoxic potential.

This is a convincing and interesting study which shows that mutated nascent chains on mitochondrial ribosomes can lead to proteotoxicity and that the proteases of the inner membrane are able to suppress these toxic effects to a certain degree. This is consistent with studies on yeast mitochondria but certainly goes far beyond previous observations. In a second part of the study, the authors report on the morphological alterations of the inner membrane that are induced by ATP6 mutations.

Specific points

1. To exclude that lower levels of mitochondrial genomes lead to the reduction in mitochondrial ribosomes and thus to lower translation levels, the authors need to quantify mtDNA levels in the cell types and fibroblast samples used in this study.
2. Fig. 2G and H shows the relative levels of ribosomal proteins. It is important to measure the levels of mitochondrial rRNAs in these samples as these normally correlate closely with the levels of assembled ribosomes. Since mitochondrial ribosomal proteins can accumulate to significant quantities in non-assembled states, their individual levels do not necessarily correlate with the amount of mitochondrial ribosomes in different samples.
3. Fig. 3B shows the stability / degradation of mt encoded proteins. However, the experiment shows that the levels of all mitochondrial translation products are much lower in the mutant. Doesn't this figure suggest that it is the synthesis of proteins that is compromised in the MT-ATP6 m.9205delTA cells and not their stability?
4. Why is the mutation in ATP6 toxic? Is this due to an impaired activity of the ATP synthase (loss of function) or due to a dominant-negative effect of this protein in the membrane (gain of function)? The introduction into the system studied here is very short. Some better explanation would certainly help here.

Point-by-point response

Reviewer #1 (Comments to the Authors (Required)):

1. Richter and colleagues examine processing of the mitochondrial protease OPA1 as an indicator of a proteotoxic stress response mediated by OMA1. Most interestingly, they convincingly demonstrate that an array of mitochondrial stressors can be alleviated by inhibiting protein synthesis from mitochondrial ribosomes. This is potentially quite interesting. Unfortunately, the authors also claim they understand the loss of mitochondrial ribosomes and the remodeling of the mitochondrial inner membrane that occurs in human disease. This is not supported.

We thank the reviewer for their interest in our research findings. However, we feel it is necessary to emphasize that our data goes far beyond the alleviation of mitochondrial stresses from proteotoxicity. Instead, we identify the origin of the major proteotoxic trigger with AFG3L2 and paraplegin dysfunction. Over the last 20 years (see the list below), disruptions to the AFG3L2 proteolytic chaperone complex were characterized at the intracellular level by the following phenotypes: (1) OMA1 activation and OPA1 processing; (2) mitochondrial membrane fragmentation; (3) remodelling of the inner mitochondrial membrane; (4) loss of mitochondrial ribosomes; and (5) a defect in oxidative phosphorylation. *But none of these studies identified the underlying trigger or mechanism.*

In our manuscript, we establish a conceptual advance by unequivocally showing the proteotoxic trigger is restricted to the nascent chain synthesis of only the 13 proteins encoded in the mitochondrial genome. Inhibiting mitochondrial protein synthesis with chloramphenicol, a well-known inhibitor of translation elongation on mitochondrial ribosomes, completely blocked all of these mitochondrial phenotypes specific to AFG3L2 and paraplegin dysfunction. We never claimed to understand the mechanisms by which mitochondrial ribosomes were degraded. Rather, we showed that proteotoxicity arising from translation of mitochondrial proteins acts as the trigger for the ribosome decay pathway. Since very little is known on the mechanisms for mitochondrial ribosome decay, investigating this research topic is clearly a separate study beyond the scope of the current manuscript.

We also demonstrate there is hierarchy to these molecular phenotypes as a part of a stress-response, arising from the absence of this mitochondrial quality control complex. In addition, we connect this mitochondrial stress response pathway to the translation of a specific class of pathogenic mitochondrial DNA mutation following heat shock, a well-known modulator of proteostasis and factor in the onset and severity of human mitochondrial disorders. The latter data may be suggestive of a type of low-level gene expression error that is routinely generated within mitochondria (e.g. aberrant mitochondrial mRNAs from imprecise processing of the polycistronic RNA as biological noise), requiring this quality control complex. Future studies will test that hypothesis.

Reference list - None of the listed studies identified the origin of the mitochondrial proteotoxicity with AFG3L2/paraplegin dysfunction.

Almajan et al. (2012) AFG3L2 supports mitochondrial protein synthesis and Purkinje cell survival. *J. Clin. Invest.* 122: 4048–4058

Casari et al. (1998) Spastic paraplegia and OXPHOS impairment caused by mutations in paraplegin, a nuclear-encoded mitochondrial metalloprotease. *Cell* 93: 973–983

Di Bella et al. (2010) Mutations in the mitochondrial protease gene AFG3L2 cause dominant hereditary ataxia SCA28. *Nat. Genet.* 42: 313–321

Ehse et al. (2009) Regulation of OPA1 processing and mitochondrial fusion by m-AAA protease isoenzymes and OMA1. *J. Cell Biol.* 187: 1023–1036

Ferreirinha et al. (2004) Axonal degeneration in paraplegin-deficient mice is associated with abnormal mitochondria and impairment of axonal transport. *J. Clin. Invest.* 113: 231–242

Gorman et al. (2015) Clonal expansion of secondary mitochondrial DNA deletions associated with spinocerebellar ataxia type 28. *JAMA Neurol.* 72: 106–111

Kondadi et al. (2014) Loss of the m-AAA protease subunit AFG3L2 causes mitochondrial transport defects and tau hyperphosphorylation. *EMBO J.* 33: 1011–1026

- Maltecca et al. (2012) Respiratory dysfunction by AFG3L2 deficiency causes decreased mitochondrial calcium uptake via organellar network fragmentation. *Hum. Mol. Genet.* 21: 3858–3870
- Maltecca et al. (2009) Haploinsufficiency of AFG3L2, the gene responsible for spinocerebellar ataxia type 28, causes mitochondria-mediated Purkinje cell dark degeneration. *J. Neurosci.* 29: 9244–9254
- Maltecca et al. (2008) The Mitochondrial Protease AFG3L2 Is Essential for Axonal Development. *Journal of Neuroscience* 28: 2827–2836
- Martinelli et al. (2009) Genetic interaction between the m-AAA protease isoenzymes reveals novel roles in cerebellar degeneration. *Hum. Mol. Genet.* 18: 2001–2013
- Merkwirth et al. (2012) Loss of prohibitin membrane scaffolds impairs mitochondrial architecture and leads to tau hyperphosphorylation and neurodegeneration. *PLoS Genet.* 8: e1003021
- Merkwirth et al. (2008) Prohibitins control cell proliferation and apoptosis by regulating OPA1-dependent cristae morphogenesis in mitochondria. *Genes Dev.* 22: 476–488
- Nolden et al. (2005) The m-AAA protease defective in hereditary spastic paraplegia controls ribosome assembly in mitochondria. *Cell* 123: 277–289
- Pfeffer et al. (2014) Mutations in the SPG7 gene cause chronic progressive external ophthalmoplegia through disordered mitochondrial DNA maintenance. *Brain* 137: 1323–1336
- Pierson et al. (2011) Whole-exome sequencing identifies homozygous AFG3L2 mutations in a spastic ataxia-neuropathy syndrome linked to mitochondrial m-AAA proteases. *PLoS Genet.* 7: e1002325

2. *A major problem with the manuscript is the writing/presentation of the Results. The Introduction and Discussion are very nicely written and accessible to a broad audience. The results section is NOT. It is very dense and nearly impossible to follow.*

We thank the reviewer for the compliment on the writing of the Introduction and Discussion. The Result section is condensed because of limitations to the character count and the amount of data generated to support our conclusions. Reviewers #2 and #3 do not appear to have any difficulty with the Result section. Nonetheless, we have taken the opportunity to rewrite sections of the Results where we felt the accessibility to a general audience could be enhanced. These changes to the text are in red font to facilitate identification.

3. *For example, the following is the title of one of the sections: "Mitochondrial protein synthesis required for lamellar cristae" This is not surprising.*

Three protein factors have been the predominant research focus on mitochondrial cristae ultrastructure: (1) dimerization of the F₁F₀ ATP synthase; (2) the MICOS complex; and (3) OPA1/Mgm1 (see Friedman and Nunnari, 2014 *Nature*; and Harner et al. 2016 *Elife*). Obviously, lipids also have an important role on membrane ultrastructure and are often disrupted with reverse genetic approaches that alter mitochondrial morphology. However, the role of mitochondrial protein synthesis as a major factor driving cristae formation has been completely overlooked. Clearly, assembly of the F₁F₀ ATP synthase requires mitochondrial protein synthesis, but the focus was the generation of positive curvature along the inner membranes at cristae tips (Rabl et al. 2009 *Journal of Cell Biology* and Davies et al. 2011 *PNAS*), not lamellar structures.

In the case of AFG3L2 dysfunction, our data definitely point to mitochondrial nascent chain synthesis as the major determinant for the alterations and remodeling of the membrane ultrastructure and morphology. Using multiple lines of investigation, we robustly established the membrane tethered OPA1 is not required for cristae junction formation, overturning one hypothesis in the literature. We also show models with acute proteolytic processing of OPA1 on its own does not stimulate fragmentation of the mitochondrial morphology, suggesting additional signals are required. Overall, there is no published literature that tested these hypotheses nor provides evidence comparable in scope and quality to data in our manuscript.

Reviewer #2 (Comments to the Authors (Required)):

1. *AFG3L2 forms a matrix facing AAA protease in mitochondria. In this study, Richter et al. analyzed the functional consequences of the defects in AFG3L2 in mitochondria. They carefully followed the effects of AFG3L2 knock down to reveal that translation-dependent activation of another protease OMA1 took place first, then loss of mitochondrial ribosomes, and OMA1-mediated processing of the dynamin-related GTPase OPA1 followed, and finally alteration in the inner membrane ultrastructure was seen. The observations are interesting, and raise many questions to stimulate further studies. However, the results at the moment remain more or less descriptive, and lack insight into the mechanism underlying the observed phenomena. In particular, it is not clear what caused translation-dependent toxicity when AFG3L2 was knocked down.*

We address many of the issues in our reply to Reviewer #1, point #1, so we draw the reviewer to this reply.

We would also like to emphasize the novelty of our data, linking phenotypes observed over the last 20 years with AFG3L2 dysfunction to the proteotoxicity of mitochondrial nascent chain synthesis. To say that this finding is descriptive is disingenuous when assessing the current state and supposed mechanistic insight of the field. None of the published papers on AFG3L2, paraplegin, or their prohibitin scaffold identified the trigger or proteotoxicity for any of the cell biology phenotypes nor a chronology to these events. Thus, the reviewer is asking us to provide a complete molecular resolution of the offending proteotoxic insult in a single manuscript that has escaped scientists for the last two decades! Moreover, *we are the first group to show what is an endogenous trigger for OMA1 activation,* as previous studies used ectopic exposure to poisons such as uncouplers to dissipate the membrane potential (see Ehses et al. 2009 Journal of Cell Biology; Baker et al. 2014 EMBO J).

Using a series of different human patient mutations, we identified one type of mitochondrial DNA mutations as a trigger for the same stress response pathway and ruled out other types of disruptions to nascent chain proteostasis. The current inability to edit animal mitochondrial DNA with genome editing tools prevents a more systematic analysis into the potential of specific classes of mitochondrial DNA mutations as proteotoxic triggers. Nonetheless, our findings make a substantial impact to the field, establishing mitochondrial nascent chain synthesis as the proteotoxic trigger and indicate a type of mitochondrial mRNA mutation that might be a specific trigger. A detailed investigation into the molecular basis of mitochondrial nascent chain synthesis proteotoxicity is a separate study beyond the scope of the current manuscript.

2. *Why degradation of mitochondrially translated proteins by AFG3L2 is required even under normal physiological conditions?*

Metabolic labelling experiments of mammalian mitochondrial protein synthesis (Wheeldon et al. 1974 European Journal of Biochemistry; and Cote et al. 1989 Journal of Biological Chemistry) robustly demonstrated up to 80% of nascent chain synthesis was not stable and rapidly degraded under steady state conditions. This points to a requirement for acutely responsive quality control mechanisms tuned to the synthesis of mitochondrial nascent chains. Therefore, it is not surprising disruptions to such a mechanisms trigger stress responses within mitochondria. One potential rationale is to prevent nascent chain overaccumulation in the membrane from disrupting membrane integrity. We established a similar conclusion using a small molecule peptidomimetic (Richter et al. 2015, Journal of Cell Biology). Furthermore, similar patterns of nascent chain degradation are

observed for cytoplasmic ribosomes (Duttler et al. 2013 Molecular Cell), which in this case is coupled to the ubiquitin-proteasomal system. Currently, very little is known or understood mechanistically how the 13 mitochondrial nascent chains are marked and recognized for degradation.

3. *To address this point, the authors could monitor the fate of translation products of mitochondrially encoded proteins, probably by RI labeling (under the condition where cytosolic protein synthesis is inhibited), with and without AFG3L2 knockdown.*

The reviewer makes a great suggestion! However, we draw the Reviewer's attention to one of our published studies (Richter et al. 2015, Journal of Cell Biology), where we quantitatively investigated the direct consequence and fate of mitochondrial nascent chains with AFG3L2 knockdown. We refer to this study in the Introduction where we describe the fate of newly synthesized mitochondrial proteins with AFG3L2 dysfunction.

4. *It is not clear, either, why mitochondrial ribosome levels decreased when AFG3L2 was knocked down; clearly activation of Oma1 facing the intermembrane space is not directly involved in this reduction of mitochondrial ribosome proteins in the matrix. Without mechanistic insight into these processes, the present study stays too premature.*

The consequence of AFG3L2 dysfunction on mitochondrial ribosomes and translation were best characterized in a mouse model (Almajan et al. 2012 Journal of Clinical Investigation) even though an oxidative phosphorylation defect was first reported in the earliest study (Casari et al. 1998 Cell). However, it is worth noting that the Almajan et al. study, nor subsequent ones, elucidated a mechanism or provide an explanation for the mitochondrial translation phenotype: "*Whether AFG3L2 is promoting ribosome formation or stability is at present still unclear.*" (Almajan et al. 2012, pg.4055). Here, we make a breakthrough and show that the effect on mitochondrial ribosomes is indirect and arises as a response from the progressive loss of quality control on mitochondrial nascent chain synthesis possibly as part of a feedback response. *Chloramphenicol completely inhibited this ribosome degradation pathway.* Moreover, we connect this mitochondrial ribosome decay pathway to the translation of a specific type of mitochondrial mRNA mutation during heat shock. Identification of the mechanisms for a mitochondrial ribosome decay is clearly a separate study well beyond the scope of the current manuscript.

5. *Fig. 1G - The asterisk indicates a wrong band for AFG3L2?*

We thank the reviewer for noting the error. The asterisk became ungrouped from the rest of the figure during preparation. The mistake has been corrected.

6. *Fig. S1A - The authors observed that prohibitin knock down caused Opa1 processing, by OMA1, but is this related to the AFG3L2-dependent OMA1 activation?*

Research from the laboratory of Thomas Langer has demonstrated prohibitins form a scaffold that physically interacts with and regulates AFG3L2 (Steglich et al. 1999 Molecular and Cellular Biology; Nolden et al. 2005 Cell). Not surprising, loss of prohibitins leads to OMA1 activation (Merkwirth et al. 2008 Genes and Development), and modulates mitochondrial protein synthesis (He et al. 2012 Nucleic Acid Research). Therefore, we wanted to test if the same mitochondrial nascent chain quality control pathway was responsible for the OMA1 activation with prohibitin knockdown. By treating the

cells with chloramphenicol we inhibited the OMA1 trigger. This findings is entirely consistent with disruptions to the quality control of mitochondrial nascent chain synthesis as the trigger for OMA1 activation as would be expected from proteins that assemble into a functional complex. Nonetheless, we have rewritten the section of Results describing these experiments to make the rationale more accessible to a general audience. The changes to the text are now in red font in the last paragraph of page 4.

7. *Fig. 2F - A control experiment to show inhibition of cytosolic translation by anisomycin is missing.*

Anisomycin is a well-known inhibitor of cytosolic protein synthesis that we routinely use at the same concentration from Fig. 2F in our mitochondrial translation assays with metabolic labeling (e.g. Richter et al. 2013 Current Biology; Richter et al. 2015 Journal of Cell Biology; Richter et al. 2018 Nature Communications). For examples in this manuscript, please see Figures 2C, 3B, S2C, S2E-F, where anisomycin addition allows us to specifically label the mitochondrial translation products. Without the inhibitor radiolabel would be incorporated into cytoplasmically translated proteins, producing a very distinct and well-characterized labeling pattern in SDS-PAGE.

Reviewer #3 (Comments to the Authors (Required)):

1. *To exclude that lower levels of mitochondrial genomes lead to the reduction in mitochondrial ribosomes and thus to lower translation levels, the authors need to quantify mtDNA levels in the cell types and fibroblast samples used in this study.*

In Figure 1, we show that the loss of mitochondrial ribosomes is not due to alterations of mtDNA abundance. In Figure 2, where we use heat shock to induce the mitochondrial ribosome decay, the pathway is triggered within 30 minutes so that after 4 hours there is less than 10% of ribosomal subunits (Figure 2G and H). The kinetics for this mitochondrial ribosomal decay is faster than the generation time determined for mitochondrial ribosome assembly (e.g. Bogenhagen et al. 2018 Cell Reports) so our observations on these acute time scales in cultured cells cannot be explained by changes to mtDNA copy number.

2. *Fig. 2G and H shows the relative levels of ribosomal proteins. It is important to measure the levels of mitochondrial rRNAs in these samples as these normally correlate closely with the levels of assembled ribosomes. Since mitochondrial ribosomal proteins can accumulate to significant quantities in non-assembled states, their individual levels do not necessarily correlate with the amount of mitochondrial ribosomes in different samples.*

We agree with the reviewer that at steady state conditions with genetic experiments disrupting mitochondrial ribosome assembly: rRNA abundance correlates with the mitochondrial ribosomal proteins. This pattern is consistently seen in human pathogenic mutations that disrupt assembly of the large or small subunit (see Suomalainen and Battersby, 2018 Nature Reviews Molecular Cell Biology). However, we disagree on the assembly status of the mitochondrial ribosomal proteins. In our sucrose gradient preparations, from human and mouse cultured cells, we consistently and robustly find the vast majority of mitochondrial ribosomal proteins are always assembled into the large and small subunits (e.g. Richter et al. 2013 Current Biology; Carroll et al. 2013 Journal of Medical Genetics; Richter et al. 2015 Journal of Cell Biology; Jackson et al. 2018 Human Molecular

Genetics). The difference in interpretation may be due to the considerable time and effort my group has spent optimising the biochemical purification of mitochondrial ribosomes. (The quality of the data in the cited references speaks for itself.)

In the experimental data of Figure 2G and 2H, heat shock triggered a decay of mitochondrial ribosomes in the MT-ATP6 m.9205delTA cells that was first preceded by the attenuation of mitochondrial protein synthesis (Figure 3B). Decay of mitochondrial ribosomes will involve both a proteolytic and RNase component, the kinetics of these responses may well differ since little is currently known on the molecular mechanisms of these pathways.

3. *Fig. 3B shows the stability / degradation of mt encoded proteins. However, the experiment shows that the levels of all mitochondrial translation products are much lower in the mutant. Doesn't this figure suggest that it is the synthesis of proteins that is compromised in the MT-ATP6 m.9205delTA cells and not their stability?*

We completely agree with the reviewer on the effect of heat shock to the synthesis of mitochondrial nascent chains in the MT-ATP6 m.9205delTA cells. In Figure 3B, the experiment tests the synthesis of the mitochondrial encoded proteins. At each time point in the heat shock incubation the cells were metabolically labeled for 30 minutes with ³⁵S-methionine/cysteine. This experiment does not measure the stability of mitochondrial nascent chains. It is worth pointing out that in the particular representative gel image of Figure 3B the baseline labeling at 37°C in the MT-ATP6 m.9205delTA mutation is lower than the wild type. Occasionally, this observation was noted but it is not a consistent phenotype as can be seen in Figure 2C. We made a mistake with the legend for Figure 3B, it was missing a title. This error has now been corrected.

To clarify the experiment and data, we rewrote the text of the manuscript and added more details to the legend of Figure 3. Changes to the text are found in the third paragraph on page 6 in red font.

4. *Why is the mutation in ATP6 toxic? Is this due to an impaired activity of the ATP synthase (loss of function) or due to a dominant-negative effect of this protein in the membrane (gain of function)? The introduction into the system studied here is very short. Some better explanation would certainly help here.*

We agree with the reviewer, it is a fascinating question why only one specific ATP6 mutation is toxic. Unfortunately, we currently do not have a mechanism for the toxicity of the MT-ATP6 m.9205delTA mutation during heat shock. At steady state, the residual ATP synthase activity and complex abundance with the m.9205delTA mutation is higher than other ATP6 mutations or with a knockdown of the F1 subunit ATP5B (See Figures S2, S3 and S4). We are currently investigating the molecular basis by which heat shock modulates the translation of the MT-ATP6 m.9205delTA mutation and generates toxicity. However, these studies are beyond the scope of the current manuscript. The only distinct feature of this mutation from all of the others we tested is the absence of a stop codon, thereby generating an aberrant mRNA that will require non-stop ribosome rescue pathways to resolve. Estimates in other cellular systems suggest that translation of aberrant mRNAs occurs at a much higher frequency than previously appreciated (please see Keiler 2015 Nature Reviews Microbiology for an excellent review on the topic). Gene expression systems have responsive RNA surveillance mechanisms to identify and resolve these mistakes. In the text, we have included more discussion on aberrant mRNAs.

December 8, 2018

Re: Life Science Alliance manuscript #LSA-2018-00219-TR

Dr. Brendan J Battersby
University of Helsinki
Institute of Biotechnology
Viikinkaari 5D
Helsinki, Uusimaa 790
Finland

Dear Dr. Battersby,

Thank you for submitting your revised manuscript entitled "A MITOCHONDRIAL STRESS RESPONSE TRIGGERED BY DEFECTS IN PROTEIN SYNTHESIS QUALITY CONTROL" to Life Science Alliance.

As pre-discussed with you, we have run the revised version only by reviewer #3 who seemed the most supportive reviewer of the manuscript upon initial review. Unfortunately, and as you can see below, reviewer #3 is not very satisfied by the revision performed and does not support publication here. We have discussed how to proceed in light of the little support provided, and have identified three essential points that need to get addressed before we can move forward with the manuscript.

The reviewer thinks that another control is needed to show that ribosome biogenesis is indeed reduced in the mutants analyzed upon heat stress, as currently the assay shows ribosome-associated proteins, not rRNA itself. While you argue that in your hands, ribosome-associated proteins are a good proxy for ribosome abundance, the reviewer insists that better support is needed. We would therefore like to ask you to either assay rRNA abundance or re-phrase your results and discussion to state that ribosome-associated proteins decline.

The reviewer thinks that the control that mtDNA levels do not decrease in the different cells analyzed is needed as currently this has been assayed upon KD of AFG3L2, but not for mt-ATP6 mutants. We would like to ask you to include such control as it seems important in light of the other results for mt-ATP6 mutants shown.

The reviewer thinks that the manuscript is a very difficult read. We agree with this view and think that this is most likely due to introduction elements and result elements being mixed, and too little background information being provided. Furthermore, the rationale to move from AFG3L2 to Mt-ATP6 is not explained well enough. We would like to ask you to heavily re-write your manuscript to make it more easily accessible to the reader.

Thank you for this interesting contribution to Life Science Alliance. We are looking forward to receiving your revised manuscript.

Sincerely,

- A letter addressing the reviewers' comments point by point.
- An editable version of the final text (.DOC or .DOCX) is needed for copyediting (no PDFs).
- High-resolution figure, supplementary figure and video files uploaded as individual files: See our detailed guidelines for preparing your production-ready images, <http://life-science-alliance.org/authorguide>
- Summary blurb (enter in submission system): A short text summarizing in a single sentence the study (max. 200 characters including spaces). This text is used in conjunction with the titles of papers, hence should be informative and complementary to the title and running title. It should describe the context and significance of the findings for a general readership; it should be written in the present tense and refer to the work in the third person. Author names should not be mentioned.

B. MANUSCRIPT ORGANIZATION AND FORMATTING:

Full guidelines are available on our Instructions for Authors page, <http://life-science-alliance.org/authorguide>

We encourage our authors to provide original source data, particularly uncropped/-processed electrophoretic blots and spreadsheets for the main figures of the manuscript. If you would like to

add source data, we would welcome one PDF/Excel-file per figure for this information. These files will be linked online as supplementary "Source Data" files.

Reviewer #3 (Comments to the Authors (Required)):

The revised version of the Richter et al study is basically identical to the initially submitted version. The authors just rephrased parts of the text and changed the order of Fig panels 2G and H. They did not address the points raised by referees 2 and 3. Though interesting, this study is still difficult to read, the data lack a number of critical controls (such as on the amount of mtDNA in the samples analysed in the different figures and a quantification of mt rRNAs). Since the data did not change, I still feel that the study needs to be carefully revised and supported by an improved set of data.

January 14, 2019

RE: Life Science Alliance Manuscript #LSA-2018-00219-TRR

Dr. Brendan J Battersby
University of Helsinki
Institute of Biotechnology
Viikinkaari 5D
Helsinki, Uusimaa 790
Finland

Dear Dr. Battersby,

Thank you for submitting your revised manuscript entitled "A MITOCHONDRIAL STRESS RESPONSE TRIGGERED BY DEFECTS IN PROTEIN SYNTHESIS QUALITY CONTROL". Please excuse the delay in getting back to you, which is due to the recent holiday season. We appreciate that you included now an analysis of mtDNA abundance in mtATP6 mutant cells as well as of ribosome abundance, and that you re-wrote your manuscript. We think that the current version is vastly improved and would thus be happy to accept your manuscript for publication here, pending final revisions to meet our journal guidelines:

- we would like to suggest slight changes to the abstract, see suggestion copied below.
- please include a 'summary blurb' in our submission system during upload of the final version
- please add callouts in the text to Fig S1D and E as well as Fig S4 B-D (other panels of these S figures are called out)
- please add callouts and legends for the movies provided

Suggestion for an altered abstract:

Mitochondria have a compartmentalized gene expression system dedicated to the synthesis of membrane proteins essential for oxidative phosphorylation. Responsive quality control mechanisms are needed to ensure that aberrant protein synthesis does not disrupt mitochondrial function. Pathogenic mutations that impede the function of the mitochondrial matrix quality control protease complex composed of AFG3L2 and Paraplegin cause a multifaceted clinical syndrome. At the cell and molecular level, defects to this quality control complex are defined by impairment to mitochondrial form and function. Here, we establish the etiology of these phenotypes. We show how disruptions to the quality control of mitochondrial protein synthesis trigger a sequential stress response characterized first by OMA1 activation followed by loss of mitochondrial ribosomes and by remodelling of mitochondrial inner membrane ultrastructure. Inhibiting mitochondrial protein synthesis with chloramphenicol completely blocks this stress response. Together, our data establish a mechanism linking major cell biological phenotypes of AFG3L2 pathogenesis and show how modulation of mitochondrial protein synthesis can exert a beneficial effect on organelle homeostasis.

You will be guided to complete the submission of your revised manuscript and to fill in all necessary

information.

A. FINAL FILES:

-- High-resolution figure, supplementary figure and video files uploaded as individual files: See our detailed guidelines for preparing your production-ready images, <http://life-science-alliance.org/authorguide>

B. MANUSCRIPT ORGANIZATION AND FORMATTING:

Full guidelines are available on our Instructions for Authors page, <http://life-science-alliance.org/authorguide>

Sincerely,

January 17, 2019

RE: Life Science Alliance Manuscript #LSA-2018-00219-TRRR

Dr. Brendan J Battersby
University of Helsinki
Institute of Biotechnology
Viikinkaari 5D
Helsinki, Uusimaa 790
Finland

Dear Dr. Battersby,

Thank you for submitting your Research Article entitled "A MITOCHONDRIAL STRESS RESPONSE TRIGGERED BY DEFECTS IN PROTEIN SYNTHESIS QUALITY CONTROL". It is a pleasure to let you know that your manuscript is now accepted for publication in Life Science Alliance. Congratulations on this interesting work.

DISTRIBUTION OF MATERIALS:

Again, congratulations on a very nice paper. I hope you found the review process to be constructive and are pleased with how the manuscript was handled editorially. We look forward to future exciting submissions from your lab.

Sincerely,

Andrea Leibfried, PhD
Executive Editor
Life Science Alliance
Meyerohofstr. 1
69117 Heidelberg, Germany
t +49 6221 8891 502
e a.leibfried@life-science-alliance.org
www.life-science-alliance.org